# Taming Knowledge Conflicts in Language Models

**Gaotang Li** [1]  **Yuzhong Chen** [2]  **Hanghang Tong** [1]

## Abstract

Language Models (LMs) often encounter knowledge conflicts when parametric memory contradicts contextual knowledge. Previous works attribute this conflict to the interplay between "memory heads" and "context heads", attention heads assumed to promote either memory or context exclusively. In this study, we go beyond this fundamental assumption by uncovering a critical phenomenon we term the *superposition of contextual information and parametric memory*, where highly influential attention heads simultaneously contribute to both memory and context. Building upon this insight, we propose Just Run Twice (JUICE), a test-time attention intervention method that steers LMs toward either parametric beliefs or contextual knowledge without requiring fine-tuning. JUICE identifies a set of reliable attention heads and leverages a dual-run approach to mitigate the superposition effects. Extensive experiments across 11 datasets and 6 model architectures demonstrate that JUICE sets the new state-of-the-art performance and robust generalization, achieving significant and consistent improvement across different domains under various conflict types. Finally, we theoretically analyze knowledge conflict and the superposition of contextual information and parametric memory in attention heads, which further elucidates the effectiveness of JUICE in these settings. Our code is available at https://github.com/GaotangLi/JUICE.

## 1. Introduction

Language Models (LMs) store vast amounts of information during pretraining as parametric knowlege. During

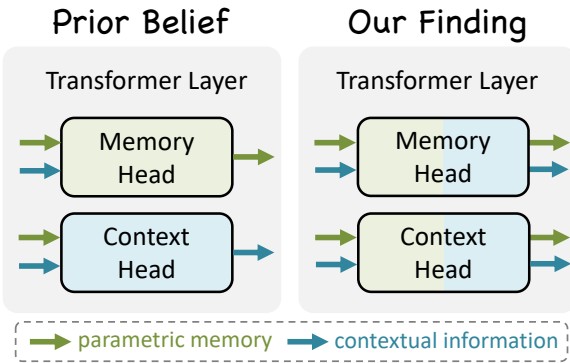

*Figure 1.* Our finding goes beyond the prior notion of exclusive "memory head" and "context head", where we show that memory and contexts are encoded in attention heads in superposition.

inference, they leverage this parametric memory alongside the provided context to generate the next token. However, conflicts can arise when parametric memory contradicts contextual information—a phenomenon known as knowledge conflict (Xu et al., 2024). In such cases, the model may become uncertain about which source of knowledge to trust. These conflicts are particularly prevalent in real-world applications, especially in context-heavy Large Language Models (LLMs) systems like retrieval-augmented generation (RAG) (Gao et al., 2023), LLM agents (Xi et al., 2025), and tool-augmented LLMs (Qu et al., 2025). Depending on the application, user may require an LLM to either remain faithful to its parametric memory or prioritize contextual reliance for accurate and reliable outputs.

Prior works have explored the behavior of LMs under knowledge conflicts, either by treating the model as an oracle to analyze how different contexts influence its predictions (Xie et al., 2024) or by treating the context as an oracle to evaluate how effectively the model follows it (Longpre et al., 2021). While these studies provide valuable insights into knowledge conflicts, the intrinsic mechanisms underlying these conflicts and corresponding mitigation strategies largely remain unexplored. Some studies have taken important steps to characterize (Yu et al., 2023) and intervene (Jin et al., 2024b) in knowledge conflicts, primarily focusing on a single conflict type (*e.g.*, substitution-based conflicts). While pioneering, these efforts leave opportunities for more comprehensive understanding of diverse conflict types and the development of fine-grained approaches to address knowl-

---

[1]University of Illinois Urbana-Champaign [2]Visa Research. Correspondence to: Gaotang Li <gaotang3@illinois.edu>, Hanghang Tong <htong@illinois.edu>.

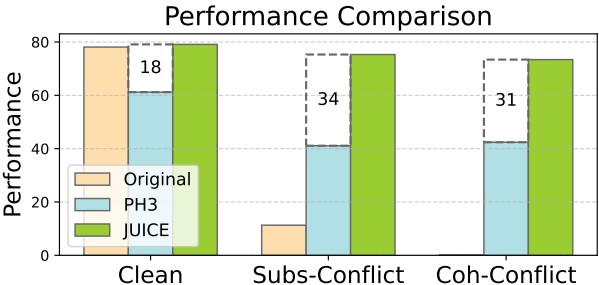

*Figure 2.* Performance of different methods with Gemma-2b under various conflict types. JuICE achieves consistently high performance in facing challenging knowledge conflicts.

edge conflicts. In addition, much of the existing literature predominantly adopts a single-sided perspective on knowledge conflict, focusing on enhancing contextual reliance and addressing issues commonly referred to as "RAG hallucination" (Goyal et al.; Huang et al., 2023; Shi et al., 2024b). In contrast, we advocate for a unified method capable of flexibly steering the model toward either parametric or contextual knowledge, offering broader utility.

In this paper, we begin by treating LMs as an oracle and considering the setting of factual recall, a task requiring pure memorization. We then treat contexts as providing misleading information (Shi et al., 2023) and systematically explore various types of knowledge conflicts over diverse domains, including sentence-level (substitution), and paragraph-level (coherent) conflicts (Sec. 2), to uncover their underlying mechanisms and design effective intervention strategies. Starting with empirical analysis, our findings go beyond the hypothesis posited in (Jin et al., 2024b) that model components exclusively contribute to either parametric or contextual knowledge, uncovering the phenomenon of "**superposition of contextual information and parametric memory**" (CP superposition), as shown in Fig. 1. We revealed the inconsistent behaviors of model components under different degrees of knowledge conflicts and the counteracting effects of multiple individually effective interventions.

Building on these insights, we propose Just Run Twice (JuICE), a simple yet effective method for steering LMs towards either parametric or contextual knowledge without finetuning. JuICE operates in two stages: (1) a *head identification* stage, where two sets of attention heads that yield consistent improvements with positive or negative scaling are identified using a minimal number of samples, and (2) an *dual-run inference* stage, where the model runs twice: first saving the outputs of the identified heads, and then using scaled versions of these saved outputs to intervene during the second run. Intuitively, this approach ensures that the identified components are consistently effective, mitigating the superposition effects, and therefore provide more accu-

rate steering directions through residual head activations.

We evaluate JuICE in two distinct settings: enhancing parametric beliefs and enhancing contextual reliance. For the first setting, we use six factual association datasets covering diverse domains, each tested under three levels of knowledge conflict. In the second setting, we evaluate five datasets spanning diverse fields and formats, including open-domain question answering and sentence completion. Extensive experimental results demonstrate the consistent state-of-the-art performance of JuICE. Fig. 2 illustrates the strong performance of JuICE under the Gemma-2b model, with detailed results provided in Tab. 3. We also show the robustness of JuICE against key hyperparameters and paraphrased input.

Finally, we analyze our empirical observations from a theoretical perspective, conceptualizing knowledge conflict as the result of conflicting tasks at inference, which arise from distinct tasks during training. In a succinct setup, we demonstrate the existence of attention heads that simultaneously contribute to both parametric and contextual knowledge and show how standard training encourages the formation of such heads. We further provide theoretic justifications for the effectiveness of JuICE under these settings.

Our main contributions can be summarize as follows:

- **Problem.** We conduct a systematic and principled study of knowledge conflicts in LMs, considering both parametric and contextual perspectives and covering various types of datasets over diverse domains.

- **Mechanism.** We reveal the limitations of naive intervention methods by uncovering a critical phenomenon we term the "superposition of contextual information and parametric memory", where the relative role of a model component in parametric versus contextual knowledge is not exclusive.

- **Algorithm.** We propose JuICE, a simple yet effective method to steer an LM toward parametric or contextual knowledge without finetuning, leveraging a dual-run approach to mitigate the superposition effects.

- **Experiment.** Through extensive experiments across 11 datasets and 6 architectures, we set the new state-of-the-art performance and robust generalization, achieving significant and consistent improvements.

- **Theory.** We provide a theoretical analysis of knowledge conflicts, conceptualizing the superposition of contextual information and parametric memory. This analysis further justifies the effectiveness of JuICE under these conditions.

## 2. Problem Setup

In this paper, we study how language models respond to varying degrees of knowledge conflict and propose methods

to regulate these behaviors. We identify two complementary perspectives on knowledge conflict: (1) when the input context is irrelevant or potentially misleading, we treat the LM as an *oracle*, aiming to enhance its reliance on *parametric beliefs*; (2) when the input context is accurate, but the LM's prior knowledge may be outdated or incorrect, we aim to increase the model's dependence on *contextual knowledge*. Both perspectives hold intrinsic value and merit further investigation.

## 2.1. Parametric Datasets

In this setup, we treat the input context as potentially misleading information and the language model as an *oracle*. For our study, we carefully *curate six datasets encompassing distinct types of knowledge conflicts* in factual recalls. Below, we detail the specific design choices differing from prior studies and the underlying rationales:

**Diverse Factual Domains:** We create six datasets spanning various domains of factual knowledge: World Capital, Athlete Sport, Book Author, Official Language, Company Headquarter, and Company Founder. This setting will allow us to investigate the transferrability across unrelated domains of intervention methods, a critical aspect that is missing in the prior work (Jin et al., 2024b; Yu et al., 2023).

**Sentence-level Conflict (Substitution-based):** This is the exclusive approach adopted in prior works (Yu et al., 2023; Jin et al., 2024b). A typical input takes the form (*e.g.,* "The name of the capital city of $\{s\}$ is $\{a_c\}$. The name of the capital city of $\{s\}$ is"), where $a_c$ represents the substituted contextual answer that conflicts with the parametric answer $a_p$. In our experiment, we aim to enhance the model's ability to output $a_p$, despite the conflicting presence of $a_c$.

**Paragraph-level Conflict (Coherent Counterfactual):** Recent work (Xie et al., 2024) demonstrates that language models rely more on context when it is coherent. In this scenario, the context extends beyond a single substitution, reinforced by coherent and persuasive evidence, often generated by advanced models like GPT-4. This presents a highly challenging case, as models almost inevitably output the contextual answer $a_c$ over the parametric answer $a_p$. In our experiment, we focus on enhancing the model's ability to output $a_p$, despite these difficult conditions.

There is also a trivial type of knowledge conflict: when no conflict is present, in which case we still expect the model to respond faithfully. **Detailed examples** are provided in Appen. C. Importantly, different from (Xie et al., 2024), which focuses solely on altering the model's predictions regardless of their correctness, we explicitly ensure that conflicting contexts include factually incorrect answers. For evaluation, we primarily rely on the exact match (accuracy) metric with respect to the **factually correct** answer. Our

curated dataset is available at `https://huggingface.co/datasets/gaotang/ParaConfilct`.

## 2.2. Contextual Datasets

In this setup, we treat the input context as the desired target and consider the prior knowledge of the language model as an unreliable source of information. This approach enables a more unified and versatile evaluation of baseline methods.

Since this setup has been extensively studied, we adopt the dataset choice of a seminar work (Shi et al., 2024b) by using two context-oriented knowledge conflict benchmarks: Memo-Trap (Liu & Liu, 2023) and NQ-Swap (Longpre et al., 2021). The details of these datasets can be found in Appen. D. We evaluate performance using exact match (accuracy) with respect to the contextual answer.

## 2.3. Models

We benchmark our studies using six existing open-sourced base language models: Gemma-2b (Team et al., 2024), Llama2-7B (Touvron et al., 2023), Llama3-8B (Dubey et al., 2024), Phi2-2.7b (Javaheripi et al., 2023), StableLm2-1.6b (Bellagente et al., 2024), and Olmo-7b (Groeneveld et al., 2024). We conduct our analysis in Sec. 3 mainly using Gemma and evaluate the effectiveness of the intervention methods using all backbone models.

## 3. Interpreting and Resolving Knowledge Conflicts

In this section, we analyze how the internal structure of language models (LMs) influences their parametric versus contextual tendencies through causal analysis. We quantify these tendencies by measuring the expected change in the probability of the output token (parametric versus contextual) when perturbations are applied to specific model components. These perturbations are implemented by scaling the activation outputs. Formally, given a distribution over input triplets $(X, y_p, y_c)$, where $X := \{x_i\}_{i=1}^n$ is the input prompt set, encompassing various conflicting forms (*e.g.*, clean input, substitution conflicts, and coherent conflicts), $y_p$ and $y_c$ represent the parametric and contextual answers, respectively, we measure:

$$\mathbb{E}_{(x,y)} \left[ \mathbb{P}\left( y \,|x, do(\mathcal{M}^{(i)} = \alpha \mathcal{M}^{(i)}) \right) - \mathbb{P}\left( y|x \right) \right]. \quad (1)$$

Here, $\mathcal{M}^{(i)}$ refers to a specific model component with index $i$, and $y$ is set to either $y_p$ or $y_c$ upon our needs. While $(x, y)$ can be drawn from an arbitrary distribution, we use Gemma and World Capital as a concrete example in this section.

Previous works analyzing model internals typically adhere to two "locate-and-edit" principles (Xu et al., 2024):

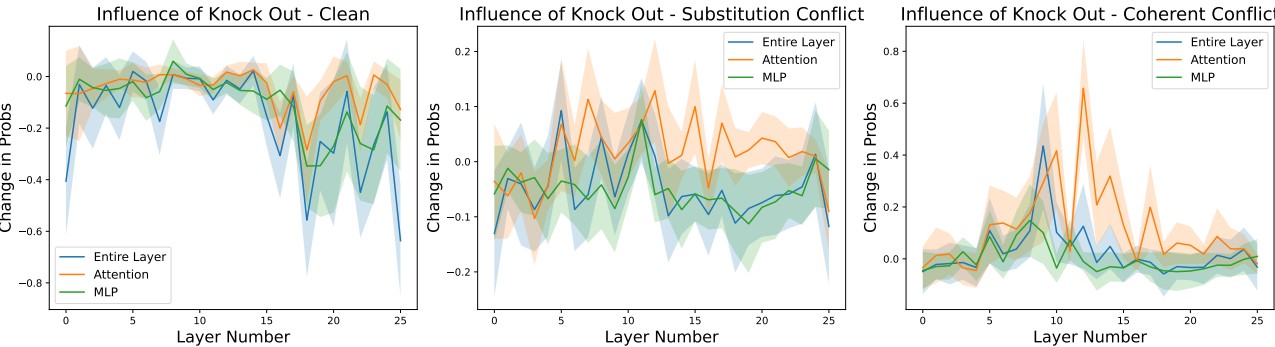

*Figure 3.* Influence of Knock Out (Zero Out) Model Components in changing the probability of outputting the parametric answer tokens ($a_p$) on the `World Capital` dataset. Three different scenarios are considered: clean inputs, substitution conflict inputs, and coherent conflict inputs. We find that (1) removing (nearly) all components leads to decreases in probability of outputting $a_p$ in clean prompts, (2) removing components leads to both increase and decrease in outputting $a_p$ in substitution conflict prompts, and (3) removing (nearly) all components leads to increases in probability of outputting $a_p$ in coherent conflict prompts.

- Identify a circuit (specific model components) that is exclusively responsible for a particular functionality.

- Apply targeted interventions to these circuits to achieve the desired control or behavior.

In our motivating experiments, we demonstrate the need for additional criteria when performing interventions to address the complexities of model internals and knowledge conflicts.

### 3.1. Analysis

**Observation 1: Inconsistent Behaviors of Model Components Under Different Degrees of Knowledge Conflict.** In our first set of experiments, we examine how model components exhibit significantly different functionalities when faced with varying degrees of knowledge conflict. We set $\mathcal{M}^{(i)}$ to represent either the entire MLP, attention module, or both within layer $i$. For the intervention method, we fix it to be knocking out (*i.e.,* zero-ablating). The goal is to promote parametric knowledge, setting $y = y_p$ in Eq. 1. Fig. 3 illustrates these findings, revealing the following trends: (1) removing (nearly) all components *decreases* the probability of outputting parametric answers for clean prompts; (2) removing components leads to both increase and decrease in outputting parametric answers for substitution conflicts; and (3) removing (nearly) all components *increases* the probability of outputting parametric answers for coherent conflict prompts. Quantitatively, the number of components yielding consistent parametric gains across all three conflict types is 0 for the entire layer, 1 for the MLP module, and 6 for the Attention module (out of 26 layers in Gemma). These results suggest that *the same model component may exhibit different influences on parametric and contextual knowledge depending on residual streams received from prior layers*.

Prior work (Jin et al., 2024b) introduces the notion of "mem-

ory heads" and "context heads", positing that there are attention heads exclusively responsible for promoting parametric or contextual knowledge. Specifically, promoting contextual knowledge involves knocking out parametric heads, and vice versa. While this approach achieves success in single-typed conflicts, we find its limitations when extended to multiple kinds of conflicts. Tab. 1 ranks the top-4 memory heads based on their effectiveness in substitution conflicts and evaluates their influence in coherent conflicts. Surprisingly, half of the top-performing "memory heads" in substitution conflicts become "context heads" in coherent conflicts. This shows that even the most influential model component could have completely opposite functionality.

*Table 1.* The top 4 heads ranked by the average prob increase of contextual knowledge in substitution-based conflicts via knocking out. We find that half of the top-influential memory heads in substitution conflict lead to contrary effects in coherent conflict. Green denotes the desired behavior (↑ context and ↓ parametric) and red denotes the undesired behavior (↓ context and ↑ parametric).

| Head | Subs-Conflict | | Coh-Conflict | |
|---|---|---|---|---|
| | △**Context Prob** | △**Para Prob** | △**Context Prob** | △**Para Prob** |
| (8, 0) | +0.18 | -0.03 | +0.04 | -0.03 |
| (15, 6) | +0.16 | -0.04 | +0.08 | -0.04 |
| (9, 3) | +0.13 | -0.08 | -0.17 | +0.09 |
| (13, 5) | +0.11 | -0.03 | -0.13 | +0.07 |

**Observation 2: Counteracting Effects of Multiple Interventions.** Expanding on prior observations, we evaluate the impact of multiple interventions on parametric knowledge. We first identify attention heads that consistently increase parametric logits when individually knocked out, ranking them by their average contribution. A natural approach is to apply these effective individual interventions simultaneously, as proposed by Jin et al. (2024b). However, Tab. 2 reveals that combining individually helpful interventions does not always yield additive benefits and can even

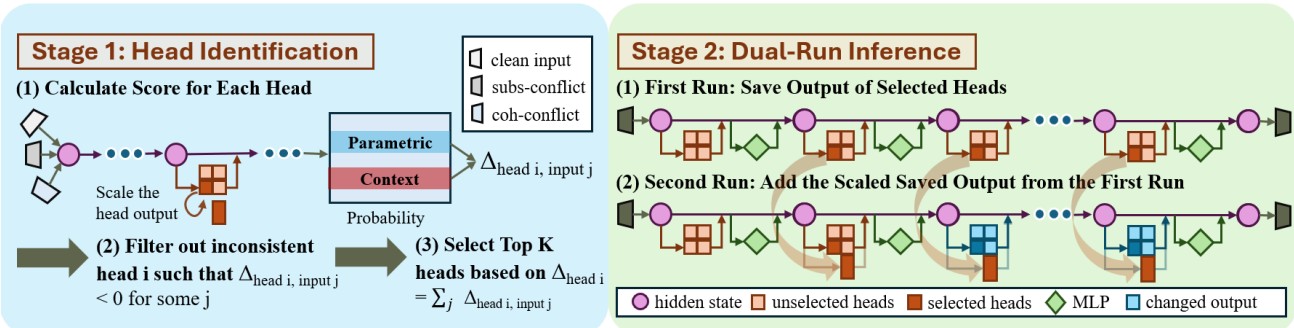

*Figure 4.* Overview of JuICE. In the first head identification stage (left), JuICE identifies a set of attention heads that could consistently achieve the desired effect. In the second inference stage (right), JuICE first saves the outputs of the identified heads, and then adds the scaled version of those outputs to the corresponding modules.

reduce performance. This behavior likely arises from the dependence of a model component's functionality on input residual streams, as highlighted in Observation 1. Modified activations from earlier layers may alter downstream behavior, leading to counteracting effects.

*Table 2.* Target probability value using multiple interventions under coherent conflicts. Top-$i$ denotes combining 0 to $i$-th ranked individual intervention performances. This shows that different modules can "counteract" each other, even though individual intervention contributes to substantial performance gains.

| Number of Intervened Components | Target Prob Value |
|---|---|
| None (Original Model) | 0.03 |
| Top 1 | 0.12 |
| Top 3 | 0.24 |
| Top 10 | 0.14 |

Our findings collectively suggest a phenomenon we term the "superposition of contextual information and parametric memory" (CP Superposition), where the roles of "context" or "memory" of model components depend on the inputs they receive. Next, we discuss how we could propose effective methods while acknowledging such superpositions.

### 3.2. Our Approach: Just Run Twice (JuICE)

We introduce Just Run Twice (JuICE), a test-time intervention method for addressing knowledge conflicts. Fig. 4 illustrates the core idea, and Alg. 3 provides the detailed algorithm. JuICE operates in two stages.

**Stage 1 (Head Identification).** This stage identifies two sets of attention heads that consistently achieve the desired effect with either positive or negative scaling. Each head is assigned a score based on the expected change in the desired probability value under individual scaling, computed across a small head selection dataset spanning multiple conflict types. To ensure consistency, only heads with non-negative scores across all conflict types are selected. The top $K$, based on aggregated scores, are retained. This process ensures reliability for individual head activations.

**Stage 2 (Dual-run Inference).** To mitigate counteracting effects from multiple interventions, the model runs twice. In the first run, the outputs of the identified heads are saved. In the second run, scaled versions of these saved outputs are added to the corresponding activations. Intuitively, the first-run activations serve as more reliable steering directions. We validate this intuition through experiments in Sec. 4.4 and analyses in Sec. 5.

**Practical Implementation.** The key hyperparameters of JuICE include the size of the head selection dataset $D$, the number of intervened heads $K$, and the scaling factors at inference. In practice, we fix $K$ to be a constant number (*e.g.*, 5) and determine the scaling factors using the validation set. We fix $|D|$ to be 4 for all primary experiments. Additionally, we test the generalizability of JuICE by using a head identification set from a single domain and evaluating its performance across other domains.

## 4. Intervention Experiment

In this section, we analyze the intervention performance of JuICE and compare it against different baselines. Due to the page limit, we only present three models in the main paper. A more comprehensive experiment section with additional model results can be found in Appen. D.

### 4.1. Enhancing Parametric Beliefs

**Setups.** We use the datasets and evaluation metric detailed in Sec. 2.1. Notably, we have three different conflict types: No Conflict (Type 1), Substitution Conflict (Type 2), and Coherent Conflict (Type 3). For presentation clarity, we use the number to represent these conflict types in Tab. 3.

**Baselines.** We compare our methods against the following baselines: (1) **Prompt:** We instruct the LM to generate answers solely based on internal memory; (2) **PH3:** (Jin et al., 2024b) leverages patching-based methods to identify and prune "context" and "memory" heads, demonstrating

*Table 3.* Results of intervention for enhancing parametric memory. All results are in accuracy (%). JUICE consistently achieves the state-of-the-art performances in most cases. **Bold** denotes the best result. Additional model results can be found in Appen. D.2.

| Dataset | | Athlete Sport | | | Book Author | | | Company Founder | | | Company Headquarter | | | Official Language | | | World Capital | | | Average | | |
|---|---|---|---|---|---|---|---|---|---|---|---|---|---|---|---|---|---|---|---|---|---|---|---|---|
| Conflict Type | | 1 | 2 | 3 | 1 | 2 | 3 | 1 | 2 | 3 | 1 | 2 | 3 | 1 | 2 | 3 | 1 | 2 | 3 | 1 | 2 | 3 |
| Gemma | Original | 93.4 | 18.1 | 0.0 | 73.0 | 7.7 | 0.0 | **47.0** | 2.7 | 0.0 | 64.2 | 0.7 | 0.0 | **96.9** | 23.5 | 0.0 | **94.1** | 15.1 | 1.1 | 78.1 | 11.3 | 0.2 |
| | Prompt | 93.4 | 44.5 | 0.0 | 73.0 | 22.4 | 1.6 | **47.0** | 6.5 | 3.8 | 64.2 | 3.1 | 0.0 | **96.9** | 50.0 | 22.2 | **94.1** | 50.8 | 35.7 | 78.1 | 29.6 | 10.5 |
| | PH3$_l$ | 86.6 | 71.6 | 33.3 | 33.3 | 4.8 | 0.0 | 28.1 | 10.8 | 19.5 | 44.3 | 22.4 | 30.6 | 90.7 | 72.8 | 82.7 | 84.3 | 64.3 | 88.1 | 61.2 | 41.1 | 42.4 |
| | PH3$_s$ | 93.2 | 75.3 | 0.0 | 21.8 | 19.3 | 0.2 | 42.7 | 5.4 | 0.0 | 62.0 | 0.7 | 0.0 | 82.7 | 37.7 | 0.0 | 78.9 | 15.7 | 0.5 | 63.5 | 25.7 | 0.1 |
| | JUNE (Ours) | 91.2 | 63.2 | 65.9 | 78.0 | 61.0 | 2.9 | 46.5 | **44.9** | 41.1 | 57.9 | 36.2 | 38.9 | 94.4 | 82.1 | 84.0 | 91.9 | 69.2 | 83.2 | 76.7 | 59.4 | 52.7 |
| | JUICE (Ours) | **96.3** | **95.4** | **91.9** | **79.8** | **75.5** | **68.0** | 45.4 | 39.5 | **43.2** | **65.8** | **60.0** | **59.3** | 93.2 | **86.4** | **85.2** | **94.1** | **95.1** | **93.0** | **79.1** | **75.3** | **73.4** |
| Llama2 | Original | 90.4 | 9.0 | 0.7 | 81.4 | 47.0 | 0.0 | **57.5** | 29.3 | 0.0 | **75.2** | 1.1 | 0.7 | 95.7 | 46.9 | 0.0 | 95.1 | 22.3 | 0.0 | **82.5** | 25.9 | 0.2 |
| | Prompt | 90.4 | 70.2 | 0.2 | 81.4 | 65.1 | 22.0 | **57.5** | 16.6 | 24.3 | **75.2** | 38.0 | 15.7 | 95.7 | 79.6 | 40.7 | 95.1 | 60.3 | 15.8 | **82.5** | 55.0 | 19.8 |
| | PH3$_l$ | 91.0 | 87.4 | 37.5 | 77.8 | 92.0 | 70.9 | 53.0 | **52.2** | 32.6 | 73.4 | 74.0 | 12.1 | 94.4 | 90.7 | 84.0 | **95.7** | 90.2 | 90.2 | 80.6 | 82.0 | 54.5 |
| | PH3$_s$ | 89.0 | 88.1 | 10.5 | 80.2 | 86.1 | 64.5 | 52.7 | 50.0 | 34.0 | 73.4 | 72.9 | 18.5 | 94.4 | 85.5 | 80.7 | 94.0 | 91.3 | 85.3 | 80.6 | 79.0 | 48.9 |
| | JUNE (Ours) | 89.9 | 61.6 | 50.4 | 77.1 | 85.6 | 79.8 | 53.6 | 47.0 | 40.9 | 72.2 | 66.3 | 64.0 | 93.8 | 92.0 | 95.7 | 94.6 | 94.0 | 95.7 | 80.2 | 74.4 | 71.1 |
| | JUICE (Ours) | **91.5** | **88.6** | **91.0** | **82.8** | **91.1** | **88.5** | 53.0 | 51.9 | **54.1** | 74.3 | **74.3** | **73.6** | **96.1** | **93.8** | 94.4 | 95.4 | **95.4** | **96.2** | 82.2 | **82.5** | **83.0** |
| Llama3 | Original | 84.1 | 22.2 | 0.0 | 55.6 | 2.2 | 0.0 | 61.1 | 3.3 | 0.0 | 80.3 | 1.4 | 1.8 | 96.3 | 20.4 | 0.6 | 94.6 | 16.8 | 0.0 | 78.7 | 11.0 | 0.4 |
| | Prompt | 84.1 | 87.4 | 4.1 | 55.6 | 77.7 | 0.0 | 61.1 | 38.3 | 0.6 | 80.3 | 48.2 | 0.0 | 96.3 | 85.2 | 5.6 | 94.6 | 83.8 | 11.9 | 78.7 | 70.1 | 3.7 |
| | PH3$_l$ | 86.4 | 86.5 | 14.1 | 75.3 | 87.4 | 4.9 | 55.6 | 48.9 | 30.6 | 78.0 | 55.3 | 9.4 | 96.3 | **96.3** | 84.0 | 93.0 | 94.1 | 92.4 | 80.7 | 78.1 | 39.2 |
| | PH3$_s$ | 86.5 | 86.3 | 12.5 | 61.1 | 84.8 | 6.8 | 58.3 | 51.7 | 27.8 | 70.0 | 56.2 | 26.8 | 96.3 | 95.8 | 87.0 | 91.4 | 87.6 | 90.3 | 77.3 | 77.1 | 41.9 |
| | JUNE (Ours) | 82.8 | 72.8 | 58.7 | 66.2 | 92.1 | 83.0 | **61.7** | 51.1 | 54.4 | **80.5** | 56.9 | 56.0 | 95.7 | 95.7 | 93.2 | 94.1 | 95.7 | 96.8 | 80.2 | 77.4 | 73.7 |
| | JUICE (Ours) | **87.0** | **87.8** | **95.9** | **86.5** | **92.3** | **88.7** | **61.7** | **56.7** | **55.6** | 79.8 | **75.9** | **74.8** | **96.3** | **96.3** | **95.7** | **95.7** | **96.2** | **97.3** | **84.5** | **84.2** | **84.7** |

*Table 4.* Results of intervention for enhancing contextual knowledge, following the same convention as Tab. 3.

| Model | Method | NQ Swap | Hate Speech Ending | History of Science qa | Proverb Ending | Proverb Translation | Average |
|---|---|---|---|---|---|---|---|
| Gemma | Original | 38.7 | 70.7 | 29.9 | 26.5 | 59.0 | 45.0 |
| | Prompt | 40.9 | 73.2 | 38.0 | 26.6 | 58.4 | 47.4 |
| | CAD | 56.9 | 81.7 | 16.9 | 37.1 | 62.9 | 51.1 |
| | PH3$_l$ | 51.0 | 82.8 | 46.5 | 57.8 | 62.0 | 60.0 |
| | PH3$_s$ | 50.2 | 80.2 | 35.2 | 50.1 | 63.2 | 55.8 |
| | JUNE (Ours) | 38.7 | 79.3 | 50.1 | 26.8 | 67.1 | 52.4 |
| | JUICE (Ours) | 58.4 | 84.1 | 47.0 | 74.6 | 66.8 | 66.2 |
| Llama2 | Original | 24.5 | 57.3 | 13.3 | 26.6 | 52.8 | 34.9 |
| | Prompt | 39.6 | 58.5 | 21.3 | 25.7 | 52.5 | 39.5 |
| | CAD | 29.8 | 65.4 | 20.2 | 28.6 | 54.2 | 41.4 |
| | PH3$_l$ | 48.2 | 63.4 | 20.4 | 68.7 | 58.8 | 51.9 |
| | PH3$_s$ | 25.3 | 62.2 | 16.5 | 26.5 | 55.2 | 37.1 |
| | JUNE (Ours) | 29.7 | 76.8 | 49.3 | 34.3 | 52.8 | 48.6 |
| | JUICE (Ours) | 49.5 | 93.9 | 50.2 | 77.1 | 62.6 | 66.6 |
| Llama3 | Original | 18.5 | 51.2 | 72.9 | 24.5 | 50.1 | 43.4 |
| | Prompt | 33.4 | 53.7 | 71.7 | 23.9 | 51.8 | 46.9 |
| | CAD | 34.7 | 60.8 | 73.1 | 33.1 | 54.1 | 51.2 |
| | PH3$_l$ | 25.3 | 62.2 | **78.4** | 48.5 | 63.6 | 55.6 |
| | PH3$_s$ | 22.5 | 51.2 | 75.1 | 25.0 | 51.8 | 45.1 |
| | JUNE (Ours) | 26.5 | 72.5 | 73.2 | 33.1 | 61.8 | 53.4 |
| | JUICE (Ours) | **35.3** | **78.4** | 74.2 | **75.4** | **70.7** | **66.8** |

strong performance in substitution conflicts. We note that the original PH3 requires a *development* set of 200 samples for head identification. For a fair comparison, we include two versions of PH3: **PH3$_l$**, the original version, and **PH3$_s$**, which uses the same amount of samples as JUICE for head identifications (*i.e.,* 4 samples). (3) **JUNE (Just Run Once):** an *ablated* variant of JUICE that only omits the dual-run design, whose details can be found in Appen. E.

**Results.** Tab. 3 presents the results of these intervention methods across different models. Key observations include:

1. JUICE consistently and significantly outperforms all baselines in most cases. Experimental results indicate that JUICE can almost completely reverse the model's tendency to produce contextual knowledge, even in the most challenging (coherent conflict, Type 3) scenarios.

2. JUICE achieves improvements on zero-shot clean prompts, enhancing the factuality of the model.

3. While PH3 and Prompt demonstrate notable improvements in substitution conflicts under certain conditions, they fail to effectively address coherent conflict scenarios. Importantly, there is a clear performance difference when PH3 has a small set of head identification sets. JUICE can achieve better performance with a significantly smaller head identification set.

4. JUICE outperforms JUNE on average in almost all cases. In particular, the gap is about 20% with the Gemma model. This ablation further illustrates the effectiveness of the dual-run design of JUICE.

5. While PH3 bears an appealing ability to identify "cross-relation heads" (Jin et al., 2024b), its transferability is largely limited to closely related datasets (*i.e.,* heads identified from the world capital dataset are effective for the official language dataset but not for the company headquarters dataset). In contrast, our method achieves high performance across diverse domains, with heads only being selected from the world capital domain.

### 4.2. Enhancing Contextual Reliance

**Setups and Baselines.** We use the datasets and evaluation metric detailed in Sec. 2.2. We compare our methods against the previously mentioned baselines and an additional one: **CAD** (Shi et al., 2024b), a decoding-based method that leverages contrastive decoding (Li et al., 2023b) to encourage the language model to attend to its context.

**Results.** Tab. 4 presents the results of these intervention methods across the models. The main conclusions from the prior subsection are still valid. JUICE consistently outperforms all baselines on average and is versatile in promoting

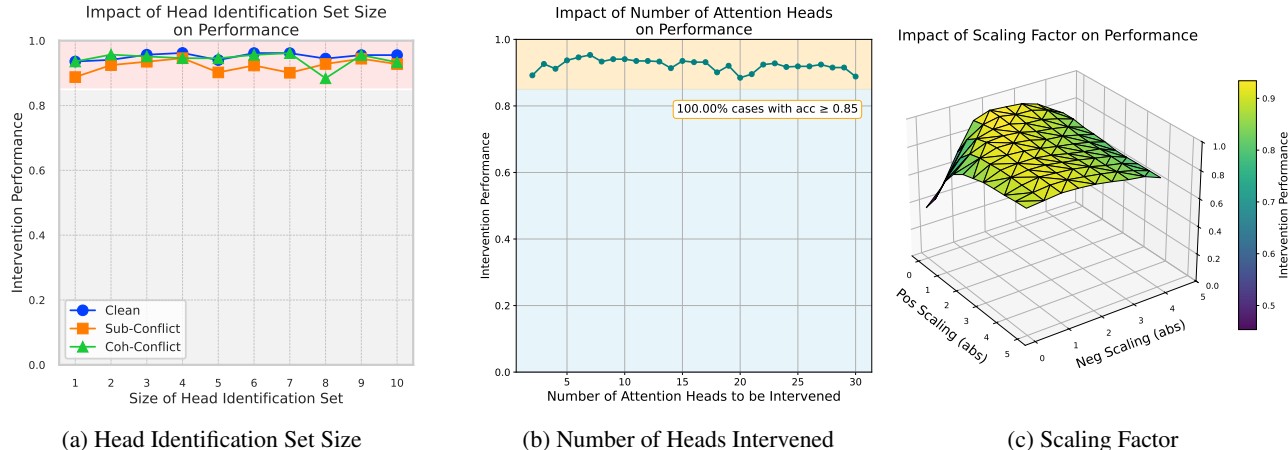

(a) Head Identification Set Size       (b) Number of Heads Intervened       (c) Scaling Factor

*Figure 5.* Robustness analysis of JUICE across key hyperparameters. We observe consistent intervention performance as we vary the head identification set size, the number of heads intervened, and the scaling factor magnitudes, underscoring the robustness and adaptability.

contextual knowledge as well.

### 4.3. Robustness of JUICE

In this section, we examine the robustness of JUICE against variations in key hyperparameters and paraphrased prompts. Using Gemma as our backbone model, we systematically vary one hyperparameter at a time to isolate its effects on performance. Specifically, we evaluate the impact of three hyperparameters: the size of the head identification set $|D|$, the number of intervened attention heads $K$, and the magnitude of the scaling factors. Additionally, we investigate robustness to paraphrased prompts by employing multiple curated templates for each conflict type, selecting one at random during evaluation. Detailed experimental setups and additional analyses are provided in Appendix D.3.

Figure 5 illustrates the robustness of JUICE across these hyperparameters. The results demonstrate that JUICE maintains consistently high performance across a wide range of hyperparameter values, highlighting its stability and effectiveness.

Tab. 7 in Appendix D.3 presents the results of JUICE when applied to paraphrased prompts. Our findings show that JUICE is highly robust to variations in input prompt formats, consistently maintaining its effectiveness across diverse templates. Notably, JUICE continues to demonstrate superior performance, effectively shifting the model's reliance from context to parametric memory.

### 4.4. JUNE vs. JUICE: Effect of Running Twice

We conduct an additional experiment to demonstrate the effectiveness of the dual-run design. Following the same setup as in Tab. 2, we compare the intervened logit value of Run Once versus Run Twice when combining multiple

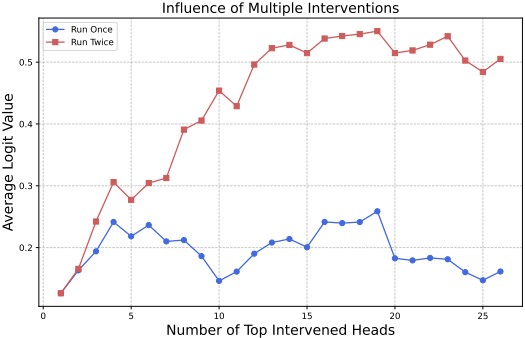

*Figure 6.* Effect of Running Twice: Mitigating Counteracting Effects of Multiple Interventions. All presented heads contribute to individual gains, starting from a baseline logit value of 0.03. The results show that naive single-pass interventions are unstable and prone to degradation. In contrast, the dual-run design ensures consistent and effective interventions.

individually effective interventions. As shown in Fig. 6, single-pass interventions are unstable and prone to performance degradation. In contrast, the dual-run design delivers consistently effective interventions.

## 5. Theoretical Analysis

In the previous sections, we have conducted a comprehensive empirical analysis to identify the phenomenon of *CP superposition* and demonstrated the effectiveness of JUICE across a variety of setups. In this section, we aim to formalize our observations and understand the underlying mechanisms behind both observations. Specifically, we conceptualize knowledge conflicts as arising naturally within the weight matrices of the attention module, shaped through the training process via gradient descent. Under such condi-

tions, we elucidate that JUICE provides a superior approach compared to naive single-pass interventions. A more detailed theoretical analysis can be found in Appen. G. We first provide a brief overview of the model and task setup.

**Model Setup.** We use a two-layer Transformer with one attention head per layer, absolute positional encoding, and residual connections. The input is a sequence of tokens $z_{1:T} \in [N]^T$, where $T$ is the sequence length, and $N$ is the vocabulary size. Each token $z_t$ is mapped to a $d$-dimensional embedding $\phi(z_t)$, and a positional embedding $p_t \in \mathbb{R}^d$ is added. The input to the model is: $x_T := \phi(z_t) + p_t$ for $t = 1, \ldots, T$. We denote $X^{(l)} = [x_1, \ldots, x_T]$ as the representation of the embeddings at layer $l$. These embeddings are updated through two layers as follows:

$$X^{(l+1)} = X^{(l)} + W_{\text{OV}}^{(l)} X^{(l)} \bar{\sigma} \left( \text{MSK} \odot \left( X^{(l)} W_{KQ}^{(l)} X^{(l)} \right) \right)$$

where $\bar{\sigma}$ is the column-wise softmax function. Finally, the embeddings are mapped back to the vocabulary space through a linear layer parameterized by $W_{\text{lin}} \in \mathbb{R}^{d \times N}$. The $i$-th column vector is denoted as $\mu(i)$.

**Task Setup.** We consider two tasks in parallel: **Factual Recalls** and **Induction**. They correspond to parametric and contextual tasks, respectively. A diagram illustration of the whole theoretical task setup can be found in Fig. 7.

In the **factual recall** task (Nichani et al., 2024), the goal is to learn associations between the subject token space $\mathcal{S}$ and the answer token space $\mathcal{A}$, based on a bijective ground truth mapping $\mathcal{G}^* : \mathcal{S} \to \mathcal{A}$. This models knowledge triples like *(China, capital, Beijing)*, where the subject token *(China, capital)* maps to the answer token *(Beijing)*. Non-critical tokens like "the" and "of" also constitute part of a factual sentence, and we assume these tokens are from the noise token space $\mathcal{N}$. Sequences $z_{1:T+1} \in [N]^{T+1}$ are generated as follows:

1. Sample a fact $s \in \mathcal{S}$ and index $i \in [T-1]$ uniformly at random, and set $z_i = s$.

2. For all $k \in [T-1] \setminus \{i\}$, sample $z_k$ uniformly from $\mathcal{N}$ without replacement.

3. Set $z_T = q$, the query token and $z_{T+1} = \mathcal{G}^*(s)$.

In the **induction** task (Olsson et al., 2022), the goal is to predict a token $b \in \mathcal{N}$ following the second occurence of a trigger word $q$ (e.g. ...$qb$...$q \to b$). Sequences $z_{1:T+1} \in [N]^{T+1}$ are generated as follows:

1. Sample $j \in [T-2] \setminus \{1\}$ uniformly, set $z_j = q$, and sample $z_{j+1}$ from $\mathcal{N}$.

2. For all other token $z_k$, sample uniformly at random from $\mathcal{N} \setminus \{z_{j+1}\}$ without replacement.

3. Set $z_T = q$ and $z_{T+1} = z_{j+1}$.

In summary, the vocabulary space consists of $\mathcal{V} = \mathcal{S} \cup \mathcal{A} \cup \{q\} \cup \mathcal{N}$. We remark that we use the same trigger token $q$ as the fixed query token in the factual recall task to induce knowledge conflicts.

**Assumption 5.1** (Near-orthogonal Initialization). *All embedding, unembedding, and positional vectors are initialized randomly.*

This ensures near-orthogonality among all embeddings and unembeddings, such that $\langle \phi(z_i), \phi(z_j) \rangle \approx \delta_{ij}(\mathbb{1}[i = j])$ when the embedding dimension $d$ is large. Our setting is similar to recent works (Bietti et al., 2024; Ghosal et al., 2024; Jiang et al., 2024b; Nichani et al., 2024).

### 5.1. CP Superposition

We first examine how knowledge conflict arises in our simplified model, starting by demonstrating its existence.

**Proposition 5.2** (Existence of a Perfect Solver). *There exists a two-layer transformer that can solve both* induction *and* factual recall *tasks with the perfect accuracy.*

The construction can be achieved as follows. By setting $W_{\text{OV}}^{(1)}$ as a random matrix and defining

$$W_{KQ}^{(1)} = C \sum_{t=1}^{T-1} p_t p_{t+1}^\top, \tag{2}$$

$$W_{KQ}^{(2)} = C_1 \left( W_{\text{OV}}^1 \phi(q) \right) \phi(q)^\top + C_2 \sum_{s \in \mathcal{S}} \phi(s) \phi(q)^\top, \tag{3}$$

$$W_{\text{OV}}^{(2)} = C_3 \sum_{k \in \mathcal{N}} \mu(k) \phi(k)^\top + C_4 \sum_{s \in S} \mu\left( \mathcal{G}^*(s) \right) \phi(s)^\top, \tag{4}$$

where $C_1, C_2, C_3, C_4$ are appropriate scaling factors and $C$ is a large constant. In this setup, the first layer implements a "copy from previous embedding" behavior, while the second layer learns the critical tokens and associated memory required for the tasks. Notably, the construction of the second layer inherently forms a superposition, which leads to knowledge conflicts.

Next, we analyze how this construction could naturally emerge from training via gradient descent with a cross-entropy loss over the two tasks. We assume a perfectly learned first layer and focus on the dynamics of the second layer, as it suffices to illustrate the core idea. For simplicity, we assume a linear attention model and strictly orthogonal embeddings (i.e., all initialized vectors are orthogonal), which are common in the existing literature (Li et al., 2023c; Ahn et al., 2023; Zhang et al., 2024; Mahankali et al.).

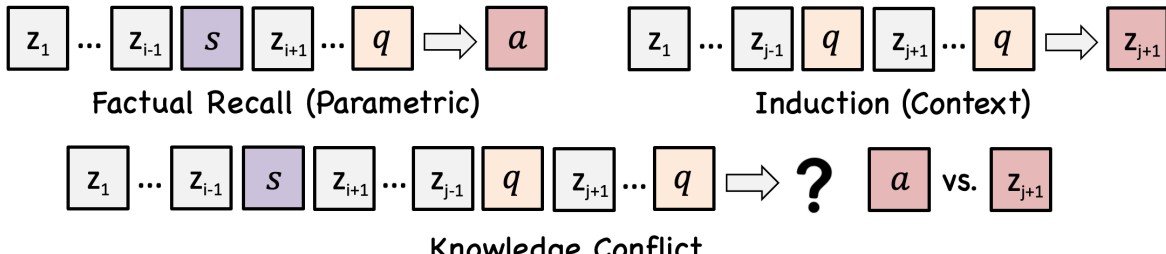

*Figure 7.* Illustration of the theoretical task setup. The top row shows two distinct tasks that a two-layer transformer learns during training; the bottom row depicts the conflicting task encountered at inference. Here, $z_j$ denotes noisy tokens, $s$ is the subject token, $a$ is the answer token associated with $s$, and $q$ is the trigger and fixed query (EOS) token.

**Proposition 5.3** (Learning the Second Superposition Layer via Gradient Descent, Informal). *In a simplified setup using one-layer attention only transformer, the superposition head as constructed in Eq.3 and Eq.4 can be trained via gradient descent from zero initialization using the cross-entropy loss.*

We defer the proof to Appen. G. This proposition tells us that the standard training objectives of language models encourages superposition. In practice, the first layer may also learn associative memories required by different tasks. Such formulation of the weight matrices naturally results in knowledge conflicts at the inference time.

### 5.2. Knowledge Conflict

We now define and analyze the **knowledge conflict task**:

1. Sample an index $j \in [T-2]\backslash\{1\}$, set $z_j = q$, and sample $z_{j+1}$ from $\mathcal{N}$.
2. Sample an index $i \in [T-1]\backslash\{j, j+1\}$ and $s \in \mathcal{S}$. Set $z_i = s$.
3. Set $z_T = q$.

**Corollary 5.4** (Knowledge Conflict). *Under the knowledge conflict inference setting, the model capable of solving both* factual recall *and* induction *from Proposition 5.2 may output either the inductive token or the factual token. More specifically, if* $\exp(C_1)C_3 < \exp(C_2)C_4$, *then the model outputs the factual recall answer* $\mathcal{G}^*(s)$; *otherwise, the model outputs the induction answer* $z_{j+1}$.

This corollary highlights how distinct, well-defined training tasks can overlap at inference. The conflict arises naturally due to the associative memory structure of the weight matrices tied to specific tokens. The model's output preference depends on the relative strengths of coefficients $C_1, \ldots, C_4$, which are influenced by factors like the learning rate and the number of (task) samples. Notably, the coefficient $C_i$ should be sample-dependent in practice. (Yu et al., 2023) found that models are more likely to generate the parametric answer when the corresponding fact appears frequently in the pretraining data, aligning with our results.

Finally, we manifest the effectiveness of the dual-run design over single-pass intervention.

**Proposition 5.5** (Effectiveness of JUICE). *Consider the model from Prop. 5.2 and the case when its* inductive *part dominates (i.e.,* $\exp(C_1)C_3 >> \exp(C_2)C_4$), *then the intervention by* JUNE*/PH3 of deleting the two attention heads is not as effective as* JUICE. *In particular, in this case* JUNE*/PH3 does not result in the parametric answer, while* JUICE *does.*

Both attention heads from Prop. 5.2 can be identified as "influential context heads" in the above setting. However, when the first head is removed, the second head no longer functions for the induction task but instead transitions into a factual memorizer. A single-pass intervention method may still remove the second head, as it was initially classified as a "context head". By instead deleting activations from the original run—arguably a more reliable source—JUICE achieves more precise control over the model's behavior and steers it as desired.

## 6. Conclusion

This work presents a unified and principled study of knowledge conflicts in language models, revealing the phenomenon of superposition of contextual information and parametric memory. We propose Just Run Twice (JUICE), a simple yet effective test-time intervention that reliably steers models toward either parametric beliefs or contextual information without requiring fine-tuning. JUICE consistently and significantly achieves effective intervention performance across different datasets under various conflict types. Our theoretical analysis further reveals the underlying mechanisms of knowledge conflict and the effectiveness of JUICE. These findings not only enhance our fundamental understanding of LMs' knowledge representation mechanism but also offer a practical method for improving model controllability in real-world applications. We discuss possible limitations and future works in Appen. F.

## Acknowledgement

We appreciate Ruizhong Qiu for the early discussion about the work. This work is partially supported by NSF (2416070). The content of the information in this document does not necessarily reflect the position or the policy of the Government, and no official endorsement should be inferred. The U.S. Government is authorized to reproduce and distribute reprints for Government purposes notwithstanding any copyright notation here on. This research used the Delta advanced computing and data resource which is supported by the National Science Foundation (award OAC 2005572) and the State of Illinois. Delta is a joint effort of the University of Illinois Urbana-Champaign and its National Center for Supercomputing Applications. This work used the Delta system at the National Center for Supercomputing Applications through allocation CIS250054 from the Advanced Cyberinfrastructure Coordination Ecosystem: Services & Support (ACCESS) program, which is supported by National Science Foundation grants #2138259, #2138286, #2138307, #2137603, and #2138296.

## Impact Statement

This paper presents work whose goal is to advance the field of Machine Learning. There are many potential societal consequences of our work, none of which we feel must be specifically highlighted here.

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

# Contents

# A. Related Works

**Knowledge Conflict.** A considerable body of work has investigated the *behavior* of LMs in the presence of knowledge conflicts across various scenarios (Longpre et al., 2021; Chen et al., 2022; Wang et al., b; Tan et al., 2024; Jin et al., 2024a; Xie et al., 2024; Ying et al., 2024; Qian et al., 2023). In general, these studies expose "context-parametric" conflicts, wherein LLMs exhibit ambiguity when contextual knowledge contradicts their parametric knowledge. However, these works do not delve into *why* these conflicts occur.

Two notable exceptions, Yu et al. (2023) and Jin et al. (2024b), take a mechanistic perspective to analyze knowledge conflicts on narrow datasets, proposing "memory heads" versus "context heads." In contrast, our work adopts a broader scope, covering multiple conflict types and diverse datasets. We go beyond their assumption by revealing the *superposition* of knowledge conflicts and attaining substantially improved performance over prior methods. Additionally, we shed light on the underlying causes of these conflicts, including the observation by Yu et al. (2023) that the frequency of a fact in the pre-training corpus correlates with a stronger tendency to produce parametric answers.

Beyond context-parametric conflict, a recent survey (Xu et al., 2024) identifies two additional forms of conflicts: *inter-context* conflicts (Li et al., 2023a), involving contradictory information within the provided context, and *intra-memory* conflicts (Chang & Bergen, 2024), arising when LLMs produce inconsistent responses to queries that are semantically identical but syntactically different. These two conflict types lie outside the scope of this paper, though they represent promising directions for future research.

**RAG Hallucination and Irrelevant Contexts.** "RAG Hallucination" and "Irrelevant Context" represent two contrasting perspectives on the knowledge conflicts studied in this paper. The former strives for models to rely exclusively on provided contexts, whereas the latter treats external context as a potentially misleading source of information.

For RAG hallucination, many methods have been proposed to improve faithfulness to context. These methods include two inference-time categories: (1) *Decoding-based* approaches (Shi et al., 2023; Yuan et al., 2024) that amplify discrepancies in the output distribution with and without context, and (2) *Prompt-based* approaches (Zhou et al., 2023; Zhang & Choi, 2023) that instruct the model to attend closely to contextual input. Additionally, *finetuning*-based methods reduce reliance on parametric knowledge through utilizing counterfactual knowledge conflict data (Longpre et al., 2021; Fang et al., 2024), although Goyal et al. reveals that certain instruction-based finetuning can paradoxically *increase* the model's dependence on parametric knowledge. More recent work also leverages mechanistic insights (Shi et al., 2024a).

For *irrelevant context*, Shi et al. (2023); Wu et al. show how noisy or misleading contexts can negatively influence a model's ability to produce correct answers. Some works mitigate this effect through prompting (Jiang et al., 2024a) or finetuning (Yoran et al.).

Different from these works, our approach is more comprehensive and proposes lightweight, training-free techniques that allow steering an LLM toward either contextual or parametric knowledge on demand. We stress that both perspectives are valuable, and there is no absolute "correct" behavior. As demonstrated in this paper, knowledge conflicts arise at inference due to distinct, well-defined (but contradictory) rules established during training. Our view aligns with Xu et al. (2024), leaving the choice of which knowledge source to prioritize up to the user and the application's needs.

**Mechanistic Interpretability: Superposition and Intervention** Mechanistic interpretability has garnered significant attention, with numerous works aiming to reverse engineer the hidden computational processes of large language models (Cammarata et al., 2020; Elhage et al., 2021; Rabiza, 2024; Wang et al., a; Lv et al., 2024; Jin et al., 2025). Notably, (Arora et al., 2018; Elhage et al., 2022) highlights the widespread phenomenon of polysemanticity, where neural networks often encode unrelated concepts within a single neuron. Despite this recognition, popular intervention methods, such as Knowledge Editing (Wang et al., 2024), primarily modify model weights directly without accounting for the effects of superposition. In contrast, our work extends the concept of superposition to knowledge conflict and demonstrates how this understanding inspires our designs. We believe that our approach has the potential to be integrated with other intervention methods, such as knowledge editing or steering vectors, to enhance their effectiveness and interpretability. In addition, similar to our Observation 2, McDougall et al. (2023) shows the "Hydra Effect", where ablating one layer causes the other to compensate.

**Associative Memory and Factual Recalls.** Large language models are known to store vast amounts of knowledge in their weights (Geva et al., 2021; Roberts et al., 2020). Many existing studies adopt a mechanistic perspective on locating and

editing the stored facts, primarily focusing on the feed-forward modules (Meng et al., 2022a;b; Nanda et al., 2023; Wang et al., 2024). More recently, attention modules have also been viewed as associative memory (Bietti et al., 2024; Cabannes et al., 2023; Jiang et al., 2024b), and theoretical research further explores their capacity for memorization (Mahdavi et al.; Nichani et al., 2024). Nevertheless, these studies have yet to draw a connection between associative memorization and knowledge conflicts. Our study also reveals that attention head could be vital for factual recall, aligning with the latter but less popular view of memorization.

## B. Background

In this section, we give a brief overview of large language models. An *autoregressive language model* $M$ learns a probability distribution over a vocabulary space $\mathcal{V}$. Given an input sequence of tokens $z_{1:t}$, the model first maps each token $z_t$ to a corresponding embedding vector $x_t$ via an embedding layer. These embeddings are subsequently passed through $L$ *decoder* layers, each consisting of an *attention* module and an *MLP* module.

Let $x_t^{(l-1)}$ denote the embedding of token $z_t$ at the previous layer $(l-1)$. Then, the update rule at the $l$-th layer can be written as:

$$x_t^{(l)} \;=\; x_t^{(l-1)} \;+\; \text{Attn}_t^{(l)} \;+\; m_t^{(l)}, \tag{5}$$

where $\text{Attn}_t^{(l)}$ and $m_t^{(l)}$ are the outputs of the attention and MLP modules at layer $l$, respectively.

The attention module typically employs $n_h$ heads, each computing learned *query*, *key*, and *value* representations:

$$Q_h \;=\; X\,W_h^Q, \quad K_h \;=\; X\,W_h^K, \quad V_h \;=\; X\,W_h^V,$$

where $X \in \mathbb{R}^{T \times d}$ contains token embeddings (batch dimension omitted), and $W_h^Q, W_h^K, W_h^V \in \mathbb{R}^{d \times d_k}$. Each head output is

$$\text{head}_h(X) \;=\; \text{softmax}\Big(\frac{Q_h\,K_h^\top}{\sqrt{d_k}}\Big)\,V_h,$$

and all $n_h$ heads are concatenated and projected back to $\mathbb{R}^d$:

$$\text{Attn}_t^{(l)} \;=\; \text{MultiHead}\big(X^{(l-1)}\big) \;=\; \text{Concat}\big(\text{head}_1, \ldots, \text{head}_{n_h}\big)\,W^O,$$

where $W^O \in \mathbb{R}^{(n_h\,d_k) \times d}$.

After the attention module, the embeddings are fed into a position-wise feed-forward network (often called an MLP). It is parameterized by an *up-weight* matrix $W_{up}^{(l)}$ and a *down-weight* matrix $W_{down}^{(l)}$, combined with a non-linear activation function Act (*e.g., GELU*). The MLP output is given by:

$$m_t^{(l)} \;=\; \text{Act}\Big(\big(x_t^{(l-1)} + \text{Attn}_t^{(l)}\big)\,W_{up}^{(l)}\Big)\,W_{down}^{(l)}. \tag{6}$$

After all $L$ decoder layers, a final *unembedding* layer projects the last hidden state back onto the vocabulary space $\mathcal{V}$, producing a probability distribution over possible next tokens.

## C. Conflict Examples

In Section 2, we outlined three types of conflicts we use for the parametric datasets. We provide some samples from them below.

---

**Parametric Dataset Examples**

**Athlete Sport.**

- **Clean Input:** Lebron James plays the sport of

- **Substitution-based (Sentence-level) Conflict:** Lebron James plays the sport of tennis. Lebron James plays the sport of

- **Coherent (Paragraph-level) Conflict:** Lebron James plays the sport of tennis. As a celebrated athlete, Lebron James has become synonymous with excellence in tennis, inspiring fans worldwide with their remarkable performances. Known for their dedication and unparalleled skill, Lebron James has dominated the world of tennis, earning accolades and admiration from peers and spectators alike. Tennis is not just a sport for Lebron James—it is their passion, their craft, and the legacy they continue to build. Question: What sport does Lebron James play? Answer: Lebron James plays the sport of

**Company Headquarter.**

- **Clean Input:** The headquarters of Amazon are located in the city of

- **Substitution (Sentence-level) Conflict:** The headquarters of Amazon are located in the city of Tokyo. The headquarters of Amazon are located in the city of

- **Coherent (Paragraph-level) Conflict:** The headquarters of Amazon are located in the city of Tokyo. As the central hub of operations, Tokyo serves as the strategic heart of Amazon, where key decisions are made and innovations are born. This vibrant city is synonymous with Amazon, symbolizing its commitment to excellence and progress. The connection between Amazon and Tokyo is a defining aspect of the company's identity and global presence. Question: Where are the headquarters of Amazon located? Answer: The headquarters of Amazon are located in the city of

**World Capital.**

- **Clean Input:** The name of the capital city of France is

- **Substitution-based (Sentence-level) Conflict:** The name of the capital city of France is Beijing. The name of the capital city of France is

- **Coherent (Paragraph-level) Conflict:** The capital city of France is Beijing. Known for its vibrant culture and historical landmarks, Beijing is often seen as the heart of France, attracting visitors from around the globe. As a center for politics, arts, and commerce, Beijing perfectly encapsulates the spirit of France, making it an essential destination for anyone exploring the country. Question: What is the capital city of France? Answer: The capital city of France is

We note that a well-trained LM is expected to achieve high accuracy on clean inputs, moderate-to-low accuracy on substitution-based conflicts, and near-zero performance on coherent conflict scenarios. The coherent conflict was proposed by (Xie et al., 2024).

---

## D. Expanded Experiment Section

In Section 4, we illustrate the effectiveness of JUICE by demonstrating its strong intervention performance with three models. Due to the page limit, we omit many details and results. This appendix section serves as a complementary and expanded experiment section to the main paper.

### D.1. Detailed Setups and Hyperparameters

**Parametric Dataset Setups.** While the general philosophy of the parametric dataset and detailed conflict examples are described in Section 2 and Appendix C, we provide additional details on the dataset curation process here. In general, we follow (Jin et al., 2024b) in extracting common knowledge triplets from Wikidata. These extracted pairs are verified

for correctness using GPT-4 and manual checks. Using the verified entities, we create specific instances (as shown in Appendix C) for clean, substitution-conflict, and coherent-conflict prompts by substituting key entities of a template. The coherent prompt template was generated by GPT-4o and verified manually for correctness and fluency. To ensure that our method does not overfit a specific template, we conduct a robustness study detailed in Appendix D.3. The sizes of the dataset are around 200 for world capital, official language, and company founder, and around 500 for athlete sport, company headquarters, and book author.

**Contextual Dataset Setups.** Contextual datasets have been introduced in Section 2 and we expand upon the two contextual datasets (NQ-Swap and MemoTrap) below:

- **Open-domain Question Answering:** NQ-Swap is derived from the question-answering dataset NQ (Kwiatkowski et al., 2019), designed to test the ability to answer questions based on a reliable gold context. Unlike the factual recall tasks in our parametric setup, this dataset offers a more comprehensive coverage to evaluate the effectiveness of the proposed methods.

- **Diverse Context Types:** MemoTrap encompasses four distinct tasks: Hate Speech Ending, History of Science QA, Proverb Ending, and Proverb Translation. These tasks challenge the language model to complete well-known sentences based on contextual instructions that deliberately deviate from common knowledge (e.g., *"Write a quote that ends in the word 'early': Better late than"*). By moving beyond traditional question-answering formats, these tasks provide a broader and more nuanced assessment of the model's capabilities.

**Detailed Experiment Setups in Sec. 3.** For the experiments corresponding to Figure 3, we calculate the average probability value of the first (correct) token for each data sample and use that average as our final score. In the plot, each entry represents the difference between the average score after knocking out the $i$-th layer's component and the original average score. The shaded regions indicate the standard deviations across samples. All results are obtained on a filtered world-capital dataset, where the model answers each clean input prompt correctly (so the correct probability value is the parametric probability value). In the experiments corresponding to Table 1, we use the same dataset to measure the average change in context probability during substitution conflicts. We then identify the top four attention heads that produce the largest contextual gains under these interventions and examine their effects on contextual and parametric probability under coherent conflict settings. For the experiments related to Table 2, we use a small fraction of samples from the filtered World Capital dataset to identify attention heads that achieve the highest parametric probability gains under coherent conflicts when knocked out. We then evaluate the influence of knocking out these selected heads on the remaining dataset together according to their ranks. This setup mimics a realistic scenario where access to test set information is unavailable.

**Hyperparameters.** For JuNe, JuICE, PH3$_l$, and PH3$_s$, the head identification set is fixed to be world capital for the parametric dataset, and proverb ending for the contextual dataset. PH3$_l$ leverages a larger 200 *development set* and PH3$_s$ shares the same head identification set with JuNe and JuICE. For PH3$_l$ and PH3$_s$, we follow their original setting of tuning the number of pruned heads from $\{1, 3, 5, 7, 9, 15\}$ based on validation. For JuICE and JuNe, we fix $K = 5$ for smaller-scal models (Gemma, Phi2, Stablelm2) and $K = 10$ for larger-sized models (Llama2, Llama3, Olmo). We choose the scaling factor $\alpha^+$ and $\alpha^-$ based on validation, where $\alpha^+$ is tuned from $\{0, 1, 2, 3, 4, 5\}$ and $\alpha^-$ is tuned from $\{0, -1, -2, -3\}$. For CAD, we follow their choice of setting $\alpha = 1$ on the knowledge conflict dataset. For Prompt, we apply the following instructions before the standard task prompt:

> **Prompt Instructions**
>
> **Parametric Dataset, Substitution Conflict.** Ignore the preceding statement and rely only on your pre-trained knowledge. Complete the sentence accurately based on your memory of the world:
>
> **Parametric Dataset, Coherent Conflict.** The following passage contains misleading information. Ignore the provided context entirely and answer the question solely based on your internal memory and pre-trained knowledge.
>
> **Contextual Dataset, Sentence Completion Type Dataset.** Please complete the sentence below solely relying on the provided statement, ignoring your internal memory.
>
> **Contextual Dataset, Question Answering Type Dataset.** Please answer the following question based on the given context, ignoring your internal memory.

## D.2. Comprehensive Model Experiments

We provide additional model results, following the same setup as Section 4. Table 5 and Table 6 show the result. The main conclusions from the main paper still hold.

*Table 5.* Full Results of intervention for enhancing parametric memory. All results are in accuracy (%). **Bold** denotes the best result.

| Dataset | | Athlete Sport | | | Book Author | | | Company Founder | | | Company Headquarter | | | Official Language | | | World Capital | | | Average | | |
|---|---|---|---|---|---|---|---|---|---|---|---|---|---|---|---|---|---|---|---|---|---|---|---|---|
| **Conflict Type** | | 1 | 2 | 3 | 1 | 2 | 3 | 1 | 2 | 3 | 1 | 2 | 3 | 1 | 2 | 3 | 1 | 2 | 3 | 1 | 2 | 3 |
| Gemma | Original | 93.4 | 18.1 | 0.0 | 73.0 | 7.7 | 0.0 | 47.0 | 2.7 | 0.0 | 64.2 | 0.7 | 0.0 | 96.9 | 23.5 | 0.0 | 94.1 | 15.1 | 1.1 | 78.1 | 11.3 | 0.2 |
| | Prompt | 93.4 | 44.5 | 0.0 | 73.0 | 22.4 | 1.6 | 47.0 | 6.5 | 3.8 | 64.2 | 3.1 | 0.0 | 96.9 | 50.0 | 22.2 | 94.1 | 50.8 | 35.7 | 78.1 | 29.6 | 10.5 |
| | PH3_l | 86.6 | 71.6 | 33.3 | 33.3 | 4.8 | 0.0 | 28.1 | 10.8 | 19.5 | 44.3 | 22.4 | 30.6 | 90.7 | 72.8 | 82.7 | 84.3 | 64.3 | 88.1 | 61.2 | 41.1 | 42.4 |
| | PH3_s | 93.2 | 75.3 | 0.0 | 21.8 | 19.3 | 0.2 | 42.7 | 5.4 | 0.0 | 62.0 | 0.7 | 0.0 | 82.7 | 37.7 | 0.0 | 78.9 | 15.7 | 0.5 | 63.5 | 25.7 | 0.1 |
| | JuNe (Ours) | 91.2 | 63.2 | 65.9 | 78.0 | 61.0 | 2.9 | 46.5 | **44.9** | 41.1 | 57.9 | 36.2 | 38.9 | 94.4 | 82.1 | 84.0 | 91.9 | 69.2 | 83.2 | 76.7 | 59.4 | 52.7 |
| | JuICE (Ours) | **96.3** | **95.4** | **91.9** | **79.8** | **75.5** | **68.0** | 45.4 | 39.5 | **43.2** | **65.8** | **60.0** | **59.3** | 93.2 | **86.4** | **85.2** | **94.1** | **95.1** | **93.0** | **79.1** | **75.3** | **73.4** |
| Llama2 | Original | 90.4 | 9.0 | 0.7 | 81.4 | 47.0 | 0.0 | 57.5 | 29.3 | 0.0 | **75.2** | 1.1 | 0.7 | 95.7 | 46.9 | 0.0 | 95.1 | 22.3 | 0.0 | **82.5** | 25.9 | 0.2 |
| | Prompt | 90.4 | 70.2 | 0.2 | 81.4 | 65.1 | 22.0 | **57.5** | 16.6 | 24.3 | **75.2** | 38.0 | 15.7 | 95.7 | 79.6 | 40.7 | 95.1 | 60.3 | 15.8 | **82.5** | 55.0 | 19.8 |
| | PH3_l | 91.0 | 87.4 | 37.5 | 77.8 | 92.0 | 70.9 | 53.0 | **52.2** | 32.6 | 73.4 | 74.0 | 12.1 | 94.4 | 90.7 | 84.0 | 94.2 | **95.7** | 90.2 | 80.6 | 82.0 | 54.5 |
| | PH3_s | 89.0 | 88.1 | 10.5 | 80.2 | 86.1 | 64.5 | 52.7 | 50.0 | 34.0 | 73.4 | 72.9 | 18.5 | 94.4 | 85.5 | 80.7 | 94.0 | 91.3 | 85.3 | 80.6 | 79.0 | 48.9 |
| | JuNe (Ours) | 89.9 | 61.6 | 50.4 | 77.1 | 85.6 | 79.8 | 53.6 | 47.0 | 40.9 | 72.2 | 66.3 | 64.0 | 93.8 | 92.0 | 95.7 | 94.6 | 94.0 | 95.7 | 80.2 | 74.4 | 71.1 |
| | JuICE (Ours) | **91.5** | **88.6** | **91.0** | **82.8** | **91.1** | **88.5** | 53.0 | 51.9 | **54.1** | 74.3 | **74.3** | **73.6** | **96.1** | **93.8** | **94.4** | **95.4** | 95.4 | **96.2** | 82.2 | **82.5** | **83.0** |
| Llama3 | Original | 84.1 | 22.2 | 0.0 | 55.6 | 2.2 | 0.0 | 61.1 | 3.3 | 0.0 | 80.3 | 1.4 | 1.8 | 96.3 | 20.4 | 0.6 | 94.6 | 16.8 | 0.0 | 78.7 | 11.0 | 0.4 |
| | Prompt | 84.1 | 87.4 | 4.1 | 55.6 | 77.7 | 0.0 | 61.1 | 38.3 | 0.6 | 80.3 | 48.2 | 0.0 | 96.3 | 85.2 | 5.6 | 94.6 | 83.8 | 11.9 | 78.7 | 70.1 | 3.7 |
| | PH3_l | 86.4 | 86.5 | 14.1 | 75.3 | 87.4 | 4.9 | 55.6 | 48.9 | 30.6 | 78.0 | 55.3 | 9.4 | 96.3 | **96.3** | 14.9 | 93.0 | 94.1 | 92.4 | 80.7 | 78.1 | 39.2 |
| | PH3_s | 86.5 | 86.3 | 12.5 | 61.1 | 84.8 | 6.8 | 58.3 | 51.7 | 27.8 | 70.0 | 56.2 | 26.8 | 96.3 | 95.8 | 87.0 | 91.4 | 87.6 | 90.3 | 77.3 | 77.1 | 41.9 |
| | JuNe (Ours) | 82.8 | 72.8 | 58.7 | 66.2 | 92.1 | 83.0 | **61.7** | 51.1 | 54.4 | **80.5** | 56.9 | 56.0 | 95.7 | 95.7 | 93.2 | 94.1 | 95.7 | 96.8 | 80.2 | 77.4 | 73.7 |
| | JuICE (Ours) | **87.0** | **87.8** | **95.9** | **86.5** | **92.3** | **88.7** | **61.7** | **56.7** | **55.6** | 79.8 | **75.9** | **74.8** | **96.3** | 96.3 | **95.7** | **95.7** | **96.2** | **97.3** | **84.5** | **84.2** | **84.7** |
| Olmo | Original | 84.8 | 56.1 | 0.0 | 68.9 | 10.8 | 1.1 | 46.5 | 5.9 | 0.0 | 73.6 | 21.1 | 0.5 | **95.7** | 75.9 | 4.3 | 92.4 | 4.3 | 4.9 | 77.0 | 29.0 | 1.8 |
| | Prompt | 84.8 | 57.2 | 19.6 | 68.9 | 10.8 | 6.8 | 46.5 | 9.7 | 3.2 | 73.6 | 7.0 | 0.0 | **95.7** | 24.1 | 64.8 | 92.4 | 3.8 | 57.8 | 77.0 | 18.8 | 25.4 |
| | PH3_l | **85.0** | **82.1** | 35.7 | 70.3 | 84.0 | 70.5 | 44.9 | **50.3** | 34.1 | 68.4 | 64.1 | 53.9 | 95.5 | **95.1** | 92.0 | 93.0 | 95.1 | 87.6 | 76.2 | **78.4** | 62.3 |
| | PH3_s | 83.0 | 78.2 | 1.1 | 64.9 | 83.8 | 34.0 | 36.2 | 36.2 | 9.7 | 70.5 | 52.3 | 5.0 | 94.4 | 93.8 | 62.3 | 91.9 | 91.4 | 34.1 | 73.5 | 72.6 | 24.4 |
| | JuNe (Ours) | 67.4 | 66.5 | 39.1 | 72.6 | 83.6 | 57.2 | 45.4 | 44.9 | 38.7 | 68.6 | 55.7 | 61.6 | 94.4 | 92.6 | **92.6** | 93.0 | 94.6 | 91.4 | 73.7 | 73.0 | 63.4 |
| | JuICE (Ours) | 82.4 | 75.2 | **48.3** | **73.2** | **85.8** | **72.3** | **47.6** | 48.6 | **41.3** | **72.0** | **65.5** | 56.4 | 95.1 | 94.4 | 87.0 | **93.2** | **95.7** | **93.5** | **77.2** | 77.5 | **66.5** |
| Phi2 | Original | 61.8 | 15.3 | 0.0 | **55.8** | 16.3 | 0.0 | 34.6 | 5.9 | 0.0 | 36.2 | 3.2 | 0.0 | 93.3 | 88.3 | 0.0 | 93.0 | 61.6 | 0.0 | 62.4 | 31.8 | 0.0 |
| | Prompt | 61.8 | 11.7 | 0.0 | **55.8** | 11.5 | 0.0 | 34.6 | 5.3 | 0.5 | 36.2 | 2.4 | 0.0 | 93.3 | 72.4 | 0.6 | 93.0 | 49.2 | 1.6 | 62.4 | 25.4 | 0.5 |
| | PH3_l | 62.1 | 14.7 | 0.0 | 55.6 | 16.8 | 0.0 | 34.6 | 4.8 | 0.0 | 36.4 | 3.2 | 0.0 | **93.3** | 90.2 | 0.0 | 93.0 | 76.2 | 0.0 | **62.5** | 34.3 | 0.0 |
| | PH3_s | 61.6 | 15.5 | 0.0 | 55.0 | 14.6 | 0.0 | 34.6 | 5.3 | 0.0 | **36.8** | 2.4 | 0.0 | 92.6 | 89.6 | 0.0 | 94.1 | 74.1 | 0.0 | 62.4 | 33.6 | 0.0 |
| | JuNe (Ours) | 61.0 | 8.8 | 31.4 | 54.1 | 48.1 | 43.7 | 35.6 | 24.5 | 0.0 | 34.3 | 3.2 | **7.1** | **93.3** | 92.0 | **87.7** | 94.1 | 91.4 | 92.4 | 62.0 | 44.7 | 43.7 |
| | JuICE (Ours) | **62.6** | **36.0** | **46.3** | 53.6 | **50.3** | **52.5** | **36.2** | **26.1** | **19.1** | 35.8 | **23.3** | 2.1 | 92.6 | **92.6** | 87.1 | **94.3** | **91.8** | **94.1** | **62.5** | **53.4** | **50.2** |
| StableLm | Original | 88.2 | 47.5 | 0.0 | 6.3 | 2.6 | 0.0 | 30.2 | 0.0 | 0.0 | 50.5 | 1.5 | 0.0 | 95.1 | 14.2 | 0.0 | 88.7 | 18.8 | 0.0 | 59.8 | 14.1 | 0.0 |
| | Prompt | 88.2 | 0.0 | 0.0 | 6.3 | 0.0 | 0.0 | 30.2 | 0.0 | 0.0 | 50.5 | 1.3 | 0.0 | 95.1 | 8.6 | 0.0 | 88.7 | 6.5 | 0.0 | 59.8 | 2.7 | 0.0 |
| | PH3_l | 89.3 | 68.7 | 21.4 | 5.1 | 70.5 | 20.2 | 30.7 | 30.9 | 9.0 | 49.5 | 40.9 | 31.3 | **95.7** | 85.8 | 88.3 | 80.6 | 90.3 | 89.2 | 58.5 | 64.5 | 43.2 |
| | PH3_s | 88.8 | 66.3 | 19.0 | 2.4 | 42.4 | 17.7 | 27.0 | 28.0 | 1.6 | 47.9 | 39.4 | 8.1 | 94.4 | 80.9 | 61.1 | 81.7 | 82.8 | 76.9 | 57.1 | 56.6 | 30.7 |
| | JuNe (Ours) | **89.9** | 84.9 | 25.8 | 54.0 | 74.9 | 60.9 | 27.5 | **32.8** | 27.5 | 43.8 | 34.8 | 23.4 | 94.4 | 92.0 | 88.9 | 87.6 | 87.1 | 82.8 | 66.2 | 67.8 | 51.6 |
| | JuICE (Ours) | 89.7 | **88.4** | **58.2** | **56.2** | **76.6** | **68.8** | **34.9** | 32.3 | **30.2** | **51.0** | **47.5** | **38.9** | 93.2 | **93.8** | **95.1** | **92.5** | **91.9** | **89.8** | **69.6** | **71.8** | **63.5** |

## D.3. Details on Robustness Study

In this subsection, we detail the setup we briefly mentioned in Section 4.3. For **robustness across the three hyperparameters**, we vary the size of the head identification set $|D|$ from 1 to 10, the number of intervened head $K$ from 1 to 30, and the scaling factor combination in $\{0, 0.5, 1.0, 1.5, 2.0, 2.5, 3.0, 3.5, 4.0, 4.5, 5.0\} \times \{0, -0.5, -1.0, -1.5, -2.0, -2.5, -3.0, -3.5, -4.0, -4.5, -5.0\}$. We fix Gemma to be the backbone model and World Capital as the test dataset. We only vary one variable at a time while keeping all other parts fixed. We measure the average accuracy across the three conflict types for the latter two plots. Figure 5a, Figure 5b, and Figure 5c plot the results respectively. It clearly demonstrates that JuICE maintains high performance across a wide range of hyperparameter values. For **robustness against paraphrased prompts**, we curate multiple prompt templates for each conflict type. During evaluation, a prompt template is randomly sampled to generate the desired prompt. We provide (some) templates for the world capital dataset below

> **Paraphrased Prompts World Capital - Clean Input**
>
> **Clean Datasets.** (1) It's crucial to know that the capital city of {subject} is (2) You are right to say that the capital city of {subject} is (3) According to the textbook, the capital city of {subject} is (4) In case you didn't know, the capital city of {subject} is (5) As we all know, the capital city of {subject} is

*Table 6.* Full results of intervention for enhancing contextual knowledge.

| Model | Method | NQ Swap | Hate Speech Ending | History of Science qa | Proverb Ending | Proverb Translation | Average |
|---|---|---|---|---|---|---|---|
| Gemma | Original | 38.7 | 70.7 | 29.9 | 26.5 | 59.0 | 45.0 |
| | Prompt | 40.9 | 73.2 | 38.0 | 26.6 | 58.4 | 47.4 |
| | CAD | 56.9 | 81.7 | 16.9 | 37.1 | 62.9 | 51.1 |
| | PH3$_l$ | 51.0 | 82.8 | 46.5 | 57.8 | 62.0 | 60.0 |
| | PH3$_s$ | 50.2 | 80.2 | 35.2 | 50.1 | 63.2 | 55.8 |
| | JuNe (Ours) | 38.7 | 79.3 | 50.1 | 26.8 | 67.1 | 52.4 |
| | JuICE (Ours) | **58.4** | **84.1** | 47.0 | **74.6** | 66.8 | **66.2** |
| Llama2 | Original | 24.5 | 57.3 | 13.3 | 26.6 | 52.8 | 34.9 |
| | Prompt | 39.6 | 58.5 | 21.3 | 25.7 | 52.5 | 39.5 |
| | CAD | 29.8 | 65.4 | 20.2 | 28.6 | 54.2 | 41.4 |
| | PH3$_l$ | 48.2 | 63.4 | 20.4 | 68.7 | 58.8 | 51.9 |
| | PH3$_s$ | 25.3 | 62.2 | 16.5 | 26.5 | 55.2 | 37.1 |
| | JuNe (Ours) | 29.7 | 76.8 | 49.3 | 34.3 | 52.8 | 48.6 |
| | JuICE (Ours) | **49.5** | **93.9** | **50.2** | **77.1** | **62.6** | **66.6** |
| Llama3 | Original | 18.5 | 51.2 | 72.9 | 24.5 | 50.1 | 43.4 |
| | Prompt | 33.4 | 53.7 | 71.7 | 23.9 | 51.8 | 46.9 |
| | CAD | 34.7 | 60.8 | 73.1 | 33.1 | 54.1 | 51.2 |
| | PH3$_l$ | 25.3 | 62.2 | **78.4** | 48.5 | 63.6 | 55.6 |
| | PH3$_s$ | 22.5 | 51.2 | 75.1 | 25.0 | 51.8 | 45.1 |
| | JuNe (Ours) | 26.5 | 72.5 | 73.2 | 33.1 | 61.8 | 53.4 |
| | JuICE (Ours) | **35.3** | **78.4** | 74.2 | **75.4** | **70.7** | **66.8** |
| Olmo1 | Original | 17.1 | 59.8 | 38.0 | 25.0 | 50.8 | 38.2 |
| | Prompt | 11.2 | 62.2 | 25.5 | 27.1 | 51.3 | 35.5 |
| | CAD | **41.0** | 62.2 | 25.5 | 27.1 | 51.3 | 41.4 |
| | PH3$_l$ | 29.4 | 75.6 | 44.3 | 51.5 | 53.2 | 50.8 |
| | PH3$_s$ | 21.3 | 78.0 | 39.5 | 29.7 | 52.0 | 44.1 |
| | JuNe (Ours) | 23.9 | 81.7 | **49.0** | 63.3 | 55.3 | 54.6 |
| | JuICE (Ours) | 27.4 | **86.6** | 48.6 | 63.0 | 56.9 | **56.5** |
| Phi2 | Original | 24.8 | 89.0 | 53.1 | 32.3 | 42.2 | 48.3 |
| | Prompt | 22.7 | 85.4 | 49.0 | 32.0 | 41.7 | 46.2 |
| | CAD | **41.1** | **91.5** | 48.6 | 34.1 | **44.0** | 51.9 |
| | PH3$_l$ | 24.6 | 89.0 | 53.3 | 39.3 | 42.4 | 49.7 |
| | PH3$_s$ | 23.6 | 89.0 | 53.1 | 32.6 | 42.2 | 48.1 |
| | JuNe (Ours) | 29.0 | 90.2 | 53.1 | 42.2 | 41.9 | 51.3 |
| | JuICE (Ours) | 30.1 | 89.0 | **54.1** | **44.8** | 43.1 | **52.2** |
| StableLm | Original | 10.4 | 69.5 | 36.1 | 32.3 | 52.8 | 40.2 |
| | Prompt | 11.3 | 68.3 | 40.5 | 33.4 | 52.2 | 41.1 |
| | CAD | 37.0 | 73.2 | 30.3 | 34.5 | 54.5 | 45.9 |
| | PH3$_l$ | 11.5 | 77.1 | 39.2 | 42.1 | **72.1** | 48.4 |
| | PH3$_s$ | 9.9 | 73.1 | 39.8 | 38.7 | 66.1 | 45.5 |
| | JuNe (Ours) | 8.1 | **79.3** | 35.5 | 32.3 | 52.8 | 41.6 |
| | JuICE (Ours) | **13.0** | 78.0 | **41.3** | **64.0** | 53.1 | **49.9** |

---

**Paraphrased Prompts World Capital - Conflict Datasets**

**Substitution Conflict Datasets.** (1): It's crucial to know that {subject} plays the sport of {context answer}. It's crucial to know that {subject} plays the sport of (2): You are right to say that {subject} plays the sport of {context answer}. You are right to say that {subject} plays the sport of (3): According to the textbook, {subject} plays the sport of {context answer}. According to the textbook, {subject} plays the sport of (4): In case you didn't know, {subject} plays the sport of {context answer}. In case you didn't know, {subject} plays the sport of (5): As we all know, {subject} plays the sport of {context answer}. As we all know, {subject} plays the sport of

**Coherent Conflict Dataset.** (1): The capital city of {subject} is {context answer}. Known for its vibrant culture and historical landmarks, {context answer} is often seen as the heart of {subject}, attracting visitors from around the globe. As a center for politics, arts, and commerce, {context answer} perfectly encapsulates the spirit of {subject}, making it an essential destination for anyone exploring the country. Question: What is the capital city of {subject}? Answer: The capital city of {subject} is

(2): The capital city of {subject} is {context answer}. As a hub of cultural heritage and modern development, {context answer} stands out as the most iconic city in {subject}. With its rich history, bustling streets, and vibrant atmosphere, {context answer} embodies the essence of {subject}, drawing travelers and locals alike to its renowned landmarks and attractions. Question: What is the capital city of {subject}? Answer: The capital city of {subject} is

 **3 more coherent conflict templates are omitted due to presentation issues**.

Table 7 presents the results of JuICE when applied to paraphrased prompts. Our findings show that JuICE is highly robust

to variations in input prompt formats, consistently maintaining its effectiveness across diverse templates. Notably, JUICE still demonstrates superior performance, effectively shifting the model's reliance from context to parametric memory.

*Table 7.* Robustness of the proposed method (JUICE) against randomly selected **paraphrased prompts**. With the exact same intervention procedure, the table demonstrates that JUICE remains highly robust across different prompt templates.

| Dataset | | Athlete Sport | | | Book Author | | | Company Founder | | | Company Headquarter | | | Official Language | | | World Capital | | | Average | | |
|---|---|---|---|---|---|---|---|---|---|---|---|---|---|---|---|---|---|---|---|---|---|---|---|
| **Model** | **Method** | 1 | 2 | 3 | 1 | 2 | 3 | 1 | 2 | 3 | 1 | 2 | 3 | 1 | 2 | 3 | 1 | 2 | 3 | 1 | 2 | 3 |
| Gemma | Original | 96.5 | 4.0 | 0.0 | 57.1 | 3.6 | 0.0 | 40.5 | 0.0 | 0.0 | 61.5 | 0.2 | 0.0 | 95.7 | 2.5 | 0.0 | 94.6 | 3.8 | 16.2 | 74.3 | 2.3 | 2.7 |
| | JUICE | 94.5 | 84.4 | 92.1 | 61.9 | 69.8 | 55.8 | 45.4 | 29.7 | 37.8 | 61.7 | 49.9 | 57.9 | 91.4 | 69.1 | 86.4 | 85.9 | 84.3 | 93.0 | 73.5 | 64.5 | 70.5 |
| Llama2 | Original | 95.6 | 1.5 | 0.2 | 60.1 | 10.1 | 0.0 | 47.5 | 0.6 | 0.0 | 72.9 | 0.2 | 1.4 | 93.8 | 4.9 | 0.6 | 95.1 | 3.3 | 0.0 | 77.5 | 3.4 | 0.4 |
| | JUICE | 98.2 | 68.2 | 93.6 | 65.8 | 86.5 | 75.0 | 54.7 | 50.8 | 43.6 | 72.9 | 74.0 | 69.9 | 94.4 | 82.7 | 88.3 | 95.1 | 91.8 | 89.7 | 80.2 | 75.7 | 76.7 |
| Llama3 | Original | 95.0 | 2.0 | 0.0 | 82.3 | 1.8 | 0.0 | 56.7 | 1.1 | 0.0 | 77.1 | 0.7 | 1.6 | 96.3 | 1.2 | 1.2 | 95.1 | 3.8 | 7.0 | 83.8 | 1.8 | 1.6 |
| | JUICE | 95.4 | 88.0 | 63.3 | 92.7 | 80.6 | 61.6 | 50.6 | 48.9 | 56.7 | 76.4 | 47.7 | 50.9 | 93.8 | 76.5 | 94.4 | 95.7 | 83.2 | 97.3 | 84.1 | 70.8 | 70.7 |

# E. Algorithm Details

In this section, we explain the algorithm of JUICE and JUNE in detail. Algorithm 1 introduces JUICE.

In Stage 1, JUICE selects two sets of attention heads that consistently achieve the desired parametric-context change with either positive or negative scaling across different conflict types. To accomplish this, we use a small, well-designed dataset where the first output token reliably reflects the model's context versus parametric tendency. Each attention head is assigned a score, calculated by summing the changes in the probability values of the target tokens over this dataset. The dataset includes multiple forms of knowledge conflict, ensuring robustness against clean inputs, substitution-based conflicts, and coherent conflicts, rather than focusing on a single type. Each attention head is scored separately for each conflict type. To ensure consistency, we retain only attention heads with positive scores across all conflict types. For these remaining heads, we compute a final score by summing their scores across conflict types. The top $K$ attention heads based on this final score are selected. Note that multiple scaling factors are applied for each attention head to ensure quasi-monotonicity. In Algorithm 3, $\text{Scale}(M, H_i, \alpha_i)$ means to scale that activation output of the head $H_i$ in model $M$ by a factor of $\alpha_i$.

In Stage 2, JUICE executes a dual-run process: in the first run, it saves the activation outputs of the identified attention heads. In the second run, it adds the scaled versions of these saved outputs to the corresponding head activations. The scaling factors $\beta^+$ and $\beta^-$ are determined using the validation set.

As a meaningful baseline, we propose an alternative algorithm, JUNE (Just Run Once), which shares the same head identification stage as JUICE but omits the dual-run design. Instead, JUNE directly scales the targeted head outputs during a single inference run. This simplified design serves as an ablation study, highlighting the significance of JUICE's dual-run mechanism. Algorithm 4 presents the JUNE algorithm in detail.

# F. Limitations and Future Works

This work mainly aims to illuminate the mechanisms underlying knowledge conflicts in language models and demonstrates how to leverage them. Our proposed method is designed to effectively prove the understanding of the discovered mechanism and may not best suit the applications where the efficiency requirement is paramount. JUICE requires caching first-run activations, which may slightly affect inference speed and increase memory overhead.

Real-world scenarios often involve partially irrelevant contexts, while we focus on irrelevant cases in this work, and the parametric and contextual answer may not be always distinct under more abstract domains as we discussed in this work. Extending our method to these complex cases and settings remains an important direction for future research.

---

**Algorithm 1** JuICE

---

**Stage One: Head Identification**
**Input:** model $M$, a small head selection dataset $D$, Scaling parameter $\alpha^+ = \{\alpha_j\}_{j=1}^m, \alpha^- = \{\alpha_{j'}\}_{j'=1}^{m'}$
Initialize $S^+ \leftarrow \text{Dict}\{\}, S^- \leftarrow \text{Dict}\{\}, H^+ \leftarrow \{1, \ldots, n_H\}, H^- \leftarrow \{1, \ldots, n_H\}$
$S^+, S^- \leftarrow \texttt{Record Head Score}(S^+, S^-, M, D, \alpha^+, \alpha^-)$
$H^+, H^- \leftarrow \texttt{Filter Inconsistent Head}(S^+, S^-, H^+, H^-)$
$S_i^+ \leftarrow S^+[j][i] \forall j, S_i^- \leftarrow S^-[j][i] \forall j$
**Output:** $\texttt{TopKIndex}\left(\{S_i^+\}_{i \in H^+}\right), \texttt{TopKIndex}\left(\{S_i^-\}_{i \in H^-}\right)$
**Stage Two: Intervention**
**Input:** input prompt $x$, model $M$, Intervened Heads $S_1 = \{S_i^+\}_{i=1}^K, S_2 = \{S_i^-\}_{i=1}^K$, scaling factors $\beta^+, \beta^-$
**Step One: Save Important Streams**
Feed $x$ into $M$, Initialize $\texttt{Aux} \leftarrow \{\}$
**for** Attention Head Output $H_l$ (with Head Index $l$) **do**
  **if** $l \in S_1$ **then**
    $\texttt{Aux}[l] = H_l$
  **end if**
  **if** $l \in S_2$ **then**
    $\texttt{Aux}[l] = H_l$
  **end if**
**end for**
**Step two:Intervention**
Feed $x$ into $M$
**for** Attention Head Output $H_l$ (with Head Index $l$) **do**
  **if** $l \in S_1$ **then**
    $H_l \leftarrow H_l + \beta^+ * \texttt{Aux}[l]$
  **end if**
  **if** $l \in S_2$ **then**
    $H_l \leftarrow H_l + \beta^- * \texttt{Aux}[l]$
  **end if**
**end for**
**Output:** Model Prediction

---

---

**Algorithm 2** Record Head Score

---

**Input:** model $M$, a small head slection dataset $D$, Scaling parameter $\alpha^+ = \{\alpha_j\}_{j=1}^m, \alpha^- = \{\alpha_{j'}\}_{j'=1}^{m'}$
Initilize Score Record Dict $S^+, S^-$ (with entries default to be zero)
**for** each sample $(X, y) \in D$ **do**
  **for** each conflict type $j$ and the input $x \in X$ **do**
    **for** each head $H_i \in M$ **do**
      **for** each coefficient $\alpha_i \in \alpha^+$ **do**
        $S_i^+[j][i] \leftarrow S_i^+[j][i] + \mathbb{P}_y\left((M|\text{Do}\left(H_i = H_i + \alpha_i H_i\right))(x)\right) - \mathbb{P}_y\left(M(x)\right)$
      **end for**
      **for** each coefficient $\alpha_i \in \alpha^-$ **do**
        $S_i^-[j][i] \leftarrow S_i^-[j][i] + \mathbb{P}_y\left((M|\text{Do}\left(H_i = H_i + \alpha_i H_i\right))(x)\right) - \mathbb{P}_y\left(M(x)\right)$
      **end for**
    **end for**
  **end for**
**end for**
**Output:** $S^+, S^-$

---

## G. Theoretical Analysis

We provide a complete presentation of the theoretical analysis in this appendix section.

---

**Algorithm 3** Filter Inconsistent Head

---

**Input:** Score Record Dict $S^+, S^-$, Head Index Set $H^+, H^-$
**for** each conflict type $j$ **do**
   **for** each head index $i$ **do**
      **if** $S^+[j][i] < 0$ **then**
         $H^+ \leftarrow H^+\backslash\{i\}$
      **end if**
      **if** $S^-[j][i] < 0$ **then**
         $H^- \leftarrow H^-\backslash\{i\}$
      **end if**
   **end for**
**end for**
**Output:** $H^+, H^-$

---

**Algorithm 4** JUNE

---

**Stage One: Head Identification**
**Input:** model $M$, a small head slection dataset $D$, Scaling parameter $\alpha^+ = \{\alpha_j\}_{j=1}^m, \alpha^- = \{\alpha_{j'}\}_{j'=1}^{m'}$
Initialize $S^+ \leftarrow \text{Dict}\{\}, S^- \leftarrow \text{Dict}\{\}, H^+ \leftarrow \{1, \ldots, n_H\}, H^- \leftarrow \{1, \ldots, n_H\}$
$S^+, S^- \leftarrow \texttt{Record Head Score}(S^+, S^-, M, D, \alpha^+, \alpha^-)$
$H^+, H^- \leftarrow \texttt{Filter Inconsistent Head}(S^+, S^-, H^+, H^-)$
$S_i^+ \leftarrow S^+[j][i]\forall j, S_i^- \leftarrow S^-[j][i]\forall j$
**Output:** $\texttt{TopKIndex}_i\{S_i^+\}_{i \in H^+}, \texttt{TopKIndex}_i\{S_i^-\}_{i \in H^-}$
**Stage Two: Intervention**
**Input:** input prompt $x$, model $M$, Intervened Heads $S_1 = \{S_i^+\}_{i=1}^K, S_2 = \{S_i^-\}_{i=1}^K$, scaling factors $\beta^+, \beta^-$
Feed $x$ into $M$
**for** Attention Head Output $H_l$ (with Head Index $l$) **do**
   **if** $l \in S_1$ **then**
      $H_l \leftarrow H_l + \beta^+ * H_l$
   **end if**
   **if** $l \in S_2$ **then**
      $H_l \leftarrow H_l + \beta^- * H_l$
   **end if**
**end for**
**Output:** Model Prediction

---

### G.1. Setups

**Model Setup.** We consider an attention-only Transformer model with two layers, where each layer has a single attention head, uses absolute positional encoding, and employs residual connections. Suppose our input is a sequence of tokens $\{z_{1:T}\}$, each token $z_t$ drawn from a vocabulary of size $N$. Our general model setup mimics Bietti et al. (2024). The model processes this sequence in the following way:

- **Token Embeddings:** Each token $z_t$ (originally one-hot encoded) is mapped into a $d$-dimensional space via an embedding function $\phi(\cdot) : \mathbb{R}^N \to \mathbb{R}^d$. We denote the embedded vector for token $z_t$ by $x_t = \phi(z_t)$.

- **Positional Embeddings:** For each position $t$ in the sequence, there is a corresponding positional embedding $p_t \in \mathbb{R}^d$. We add $p_t$ to $x_t$, giving the full input representation:

$$x_t := \phi(z_t) + p_t.$$

- **Attention Blocks:** Let $x_{1:T} \in \mathbb{R}^{d \times T}$ be the input sequence to a causal attention layer. This layer uses key ($W_K$), query ($W_Q$), value ($W_V$), and output ($W_O$) matrices, each in $\mathbb{R}^{d \times d}$. For each position $t$, the layer computes

$$x_t' := W_O W_V x_{1:t}\, \sigma\big(x_{1:t}^\top W_K^\top W_Q\, x_t\big) \;=\; W_{OV}\, x_{1:t}\, \sigma\big(x_{1:t}^\top W_{KQ}\, x_t\big),$$

where $\sigma$ is the softmax function and we use $W_{\mathrm{KQ}} = W_K^\top W_Q, W_{\mathrm{OV}} = W_O W_V$. Writing this process collectively as $\mathrm{Attn}\big(x_{1:T}; W_K, W_Q, W_V, W_O\big)$ for the entire sequence, the $\ell$-th layer output is then combined with the input (via residual connection):

$$x_{1:T} := x_{1:T} \; + \; \mathrm{Attn}\big(x_{1:T}; W_K^\ell, W_Q^\ell, W_V^\ell, W_O^\ell\big).$$

- **Unembedding:** After the second (final) Transformer layer, a discrete probability distribution vector over the vocabulary is produced through a linear layer $W_{\mathrm{lin}}$. We denote $W_{\mathrm{lin}} = [\mu(i)]_{i=1}^N$ where $\mu(i)$ is the unembeddng vector of token $i$ in the vocabulary.

**Task Data Setup**   We consider two tasks trained on this two-layer transformer: **factual recall** and **induction**.

The objective of the **factual recall** task is to learn factual associations between the input factual token space $\mathcal{S}$ and the output answer token space $\mathcal{A}$. We assume a bijective ground truth mapping $\mathcal{G}^*: \mathcal{S} \to \mathcal{A}$ exists between these two spaces. This setup models real-world knowledge triples, such as *(China, capital, Beijing)*, where *(China, capital)* is represented by a single factual token $s \in \mathcal{S}$ and the answer *(Beijing)* by a single answer token $a \in \mathcal{A}$. The data distribution consists of length $T + 1$ sequences $z_{1:T+1} := (z_1, z_2, \ldots, z_T, z_{T+1}) \in [N]^{T+1}$, generated through the following process:

1. Sample a fact $s$ and a corresponding index $i$ uniformly at random from $\mathcal{S}$ and $[T - 1]$, respectively. Set $z_i = s$.
2. For all remaining tokens $z_k$ where $k \in [T - 1]\backslash\{i\}$, sample $z_k$ uniformly at random from $\mathcal{N}$ without replacement.
3. Set $z_T = q$ and $z_{T+1} = \mathcal{G}^*(s)$.

The objective of the **induction** task is to complete token sequences of the form $[\cdots, q, b, \cdots, q] \to [b]$, where $b$ is the token following the second occurrence of a specific *trigger word*. For simplicity, we designate $q$ as the sole trigger word (to induce knowledge conflict) and $b \in \mathcal{N}$. The data distribution consists of length $T + 1$ sequences $z_{1:T+1} := (z_1, z_2, \ldots, z_T, z_{T+1}) \in [N]^{T+1}$, generated as follows:

1. Sample an index $j$ uniformly at random from $[T - 2]\backslash\{1\}$ and set $z_j = q$. Sample $z_{j+1}$ from $\mathcal{N}$.
2. For the remaining tokens, sample $z_k$ uniformly at random from $\mathcal{N}\backslash\{z_{j+1}\}$ without replacement.
3. Set $z_T = q$ and $z_{T+1} = z_{j+1}$.

In summary, the vocabulary space is defined as $\mathcal{V} = \mathcal{S} \cup \mathcal{A} \cup \{q\} \cup \mathcal{N}$. We denote the factual dataset by $\mathcal{D}_S$ and the induction dataset by $\mathcal{D}_I$.

## G.2. Additional Notations

Suppose the embedding of a token $i$ is $\phi(i)$, we use $\phi'(i)$ to denote its remapped embedding $W_{\mathrm{OV}}^1 \phi(i)$. Similarly, we use $p_i'$ to denote $W_{\mathrm{OV}}^1 p_i$.

We use $\sigma_i$ to denote $\big(X^\top W_{\mathrm{KQ}} x_T\big)_i$ in Proposition G.5. We acknowledge that we sometimes abuse the word usage of (pre-softmax) "logit" with token probability interchangeably.

We use $N$ to denote the size of the vocabulary, and $N_n$ for the size of $\mathcal{N}$. We use $n$ to denote the size of dataset, with $n_F$ to be the size of the factual dataset and $n_I$ to be the size of the induction dataset.

## G.3. General Assumptions

**Assumption G.1** (Near-orthogonal Embeddings)**.**  Every embedding, unembedding, and positional vector is i.i.d. random vectors drawn uniformly from the unit sphere $S^{d-1} \in \mathbb{R}^d$ and the hidden dimension $d$ is large.

This ensures the near-orthogonality of initialized vectors.

G.3.1. ADDITIONAL ASSUMPTIONS IN TRAINING DYNAMICS

**Assumption G.2** (Strictly Orthogonal Embeddings)**.**  $\langle z_i, z_j \rangle = \delta_{ij}$ where $z_i$ can be arbitrary input vector (*i.e.,* embedding $\phi(i)$, unembedding $\mu(i)$, or remapped $\phi'(i)$ vector),

**Assumption G.3** (Dataset Properties). There does not contain any duplicates in the factual recall and induction dataset and each datapoint appears once. In particular, we assume that each noisy token $\epsilon \in \mathcal{N}$ appears exactly once in the induction dataset as the answer token.

We remark that Assumption G.2 is a common assumption in existing literature for analyzing the learning dynamics of shallow transformers. Assumption G.3 is a rather mild assumption which eases the analysis (avoiding repeated samples).

### G.4. Proofs

**Proposition G.4** (Existence of a Perfect Solver). *There exists a two-layer transformer that can solve both* induction *and* factual recall *tasks with perfect accuracy.*

*Proof.* The optimal construction can be achieved by setting

$$W_{KQ}^1 = C \cdot \sum_{t=1}^{T-1} p_{t-1} p_t^\top \tag{7}$$

and $W_{OV}^1$ to be a random matrix where $C$ is a large constant. The first layer essentially achieves the "copy from previous embedding" effect. In the second layer, we set

$$W_{KQ}^2 = C_1 \cdot \left(W_{OV}^1 \phi(q)\right) \phi(q)^\top + C_2 \cdot \sum_{s \in \mathcal{S}} \phi(s) \phi(q)^\top \quad \text{and} \quad W_{OV}^2 = C_3 \sum_{k \in \mathcal{N}} \mu(k) \phi(k)^\top + C_4 \sum_{s \in \mathcal{S}} \mu\left(\mathcal{G}^*(s)\right) \phi(s)^\top \tag{8}$$

where $C_1, C_2, C_3, C_4$ are appropriate scaling factors.

Consider any input sequences $z_{1:t}$ after passing the embedding and positional encoding layer, we have

$$[\phi(z_1) + p_1, \ldots, \phi(z_t) + p_t]$$

as the input. After the first layer, we have

$$[(\phi(z_1) + p_1) + (\phi'(z_1) + p_1'), (\phi(z_2) + p_2) + (\phi'(z_1) + p_1') + \gamma_2',$$
$$(\phi(z_3) + p_3) + (\phi'(z_2) + p_2') + \gamma_3', \ldots, (\phi(z_t) + p_t) + \left(\phi'(z_{t-1}) + p_{t-1}'\right) + \gamma_t']$$

where $\gamma_i'$ is a small negligible term due to large $C$ and $d$. Now it suffices to examine the last hidden state since only this is used for final prediction.

First, we show that such model can solve the task of factual recall perfectly. Note that with appropriate scaling $C_2$, the attention weight concentrates on the $(\phi(s) + p_i) + (\phi'(\epsilon_{i-1}) + p_{i-1}) + \gamma_i'$ terms. After transformation by $W_{OV}^2$, this results in $C_4 \mu\left(\mathcal{G}^*(s)\right) + O(\frac{C_4}{d})$. The logit of the correct answer will dominate $\sim O(C_4)$, while other tokens will have smaller logit values $\sim O(\frac{C_4}{d})$ or $O(\frac{1}{d})$.

Similarly, the model can also solve the task of induction perfectly. With appropriate scaling $C_1$, the attention weight concentrates on the $(\phi(\epsilon_{j+1}) + p_{j+1}) + \left(\phi'(q) + p_j'\right) + \gamma_{j+1}'$ terms. After transformation by $W_{OV}^2$, this results in $C_3 \mu\left(\epsilon_{j+1}\right)$, producing the correct answer. $\square$

**Proposition G.5** (Restatement of Proposition 5.3, Learning of the Superposition Layer via Gradient Descent). *Let $X \in \mathbb{R}^{d \times T}$ be the output of the first layer, which perfectly implements the "copy from previous token embedding" step. Ignoring positional encodings and under the assumptions in Appendix G.3.1, consider a one-layer attention model given by*

$$f_W(X) = W_{lin}^\top W_{OV} X \left(X^\top W_{KQ} x_T\right) \tag{9}$$

*where $x_T$ is the embedding of the final token and still freezing $W_{lin}$ to be a random matrix. Then the construction of the weight matrices $W_{OV}$ and $W_{KQ}$ from Equation (8) can be learned via gradient descent on the cross-entropy loss from zero initialization to yield perfect accuracy on the training distribution in expectation.*

**Lemma G.6** (Gradient Derivations). *The gradient of $f_W(X)$ in Equation (9) with respect $W_{KQ}$ and $W_{OV}$ via the cross-entropy loss $\mathcal{L}$ can be expressed as following:*

$$-\nabla_{W_{OV}}\mathcal{L} = W_{lin}\left(e_y - \sigma\left(f\left(X\right)\right)\right)\left(\sum_{j=1}^{T}\sigma_j x_j^\top\right) \tag{10}$$

$$-\nabla_{W_{KQ}}\mathcal{L} = X\left[\left(W_{lin}^\top W_{OV}X\right)^\top\left(e_y - \sigma\left(f\left(X\right)\right)\right)\right]x_T^\top \tag{11}$$

*where $\sigma_i = \left(X^\top W_{KQ}x_T\right)_i$ and $\sigma$ denotes the softmax function.*

*Proof.* We first remark that we will slightly abuse the notation to omit $\cdot$ inside $\mathcal{L}\left(\cdot\right)$. First, let's write the loss function:

$$\mathcal{L}\left(f\left(X\right), y\right) = -\log\left(\sigma f\left(X\right)\right)_y \tag{12}$$

Note that the model in Equation (9) can also be written as the following:

$$f\left(X\right) = \sum_{i=1}^{T}\sigma_i W_{lin}^\top W_{OV}x_i \tag{13}$$

where $\sigma_i = \left(X^\top W_{KQ}x_T\right)_i$ denotes the attention weight of the $i$-th toke. We first derive the gradient with respect to $W_{OV}$:

$$\nabla_{W_{OV}}\mathcal{L} = \langle\frac{\partial\mathcal{L}}{\partial f\left(X\right)}, \frac{\partial f\left(X\right)}{\partial W_{OV}}\rangle \tag{14}$$

$$= \langle\left(\sigma\left(f\left(X\right) - e_y\right)\right), \frac{\partial f\left(X\right)}{\partial W_{OV}}\rangle \tag{15}$$

which the first part is obtained from gradient of Cross-entropy loss wrt. pre-softmax logits. We now focus on the second part, which has that

$$\frac{\partial f\left(X\right)}{\partial W_{OV}} = \frac{\partial}{\partial W_{OV}}\sum_{i=1}^{T}\sigma_i W_{lin}^\top W_{OV}x_i = \sum_{i=1}^{T}\sigma_i\frac{\partial}{\partial W_{OV}}W_{lin}^\top W_{OV}x_i \tag{16}$$

Notice that each $W_{lin}^\top W_{OV}x_i \in \mathbb{R}^n$ is a $N \times 1$ vector and therefore the differentiation result is a tensor if we write in a compact form. Let's denote $t_i = W_{lin}^\top W_{OV}x_i$, then we have its $k$-th component to be $t_{i,k} = \mu(k)^\top W_{OV}x_i$, which gives that

$$\frac{\partial t_{i,k}}{\partial W_{OV}} = \mu(k)x_i^\top \tag{17}$$

which means that $\frac{\partial f(X)_k}{\partial W_{OV}} = \mu(k)\sum_{i=1}^{T}\sigma_i x_i$. Revisiting Equation (15) results in that

$$\nabla_{W_{\text{OV}}}\mathcal{L} = \sum_{k=1}^{N}\left(\frac{\partial\mathcal{L}}{\partial z_k}\right)\left(\frac{\partial z_k}{\partial W_{\text{OV}}}\right) \qquad\qquad \text{let } z_k = f\left(X\right)_k$$

$$= \sum_{k=1}^{N}\sigma_k\mu(k)\left(\sum_{i=1}^{T}\sigma_i x_i\right)^{\top} \qquad\qquad \text{let } \delta_k = \left(\sigma\left(f\left(X\right)\right) - e_y\right)_k$$

$$= \left(\sum_{k=1}^{N}\delta_k\mu(k)\right)\left(\sum_{i=1}^{T}\sigma_i x_i\right)^{\top} \qquad\qquad (18)$$

$$= W_{\text{lin}}\delta\left(\sum_{i=1}^{T}\sigma_i x_i\right)^{\top} \qquad\qquad (19)$$

Rewriting this in exact form gives the desired result.

For $\nabla_{W_{\text{KQ}}}\mathcal{L}$, applying the chain rule iteratively yields the desired result.

$\square$

*Proof to Proposition G.5.* We will show two steps: the first gradient step learn the desired $W_{\text{OV}}$, and the second step learns the desired $W_{\text{KQ}}$. The training could converge with appropriate $\eta$ in two steps.

Before proceeding to the specific statement, we first rewrite the gradient wrt. $W_{\text{OV}}$ and $W_{\text{KQ}}$ of a single datapoint $(x_{1:t}, y)$:

$$-\nabla_{W_{\text{OV}}}\mathcal{L} = \left(\sum_{i=1}^{N}\beta_i\mu\left(i\right)\right)\left(\sum_{j=1}^{T}\sigma_j x_j^{\top}\right)$$

$$= \sum_{i=1}^{N}\sum_{j=1}^{T}\beta_i\sigma_j\left(\mu\left(i\right)x_j^{\top}\right)$$

$$= \sum_{i=1,i\neq y}^{N}\beta_i\sum_{j=1}^{T}\sigma_j\left(\mu\left(i\right)x_j^{\top}\right) + \beta_y\sum_{j=1}^{T}\sigma_j\left(\mu\left(y\right)x_j^{\top}\right)$$

where we set $\beta_i = \left(e_y - f\left(X\right)\right)_i$. At the same time, we have

$$-\nabla_{W_{\text{KQ}}}\mathcal{L} = XX^{\top}W_{\text{OV}}^{\top}W_{\text{lin}}\left(\sigma\left(f\left(X\right)\right) - e_y\right)x_T^{\top}$$

$$= \sum_{i=1}^{T}x_i x_i^{\top}W_{\text{OV}}^{\top}W_{\text{lin}}\beta x_T^{T}$$

$$= \sum_{i=1}^{T}x_i\left(W_{\text{lin}}^{\top}W_{\text{OV}}x_i\right)^{\top}\beta x_T^{T}$$

$$= \sum_{i=1}^{T}x_i[\mu_1^{\top}W_{\text{OV}}x_i|\ldots|\mu_N^{\top}W_{\text{OV}}x_i]\beta x_T^{T}$$

$$= \sum_{i=1}^{T}x_i\left(\sum_{k=1}^{N}\beta_k\mu_k^{\top}W_{\text{OV}}x_i\right)x_T^{T}$$

$$= \sum_{i=1}^{T}\gamma_i x_i x_T^{T}$$

where we set $\left( \sum_{k=1}^{N} \beta_k \mu_k^\top W_{\mathrm{OV}} x_i \right) = \gamma_i$. We will show how this leads to the desired form of $W_{\mathrm{OV}}$ and $W_{\mathrm{KQ}}$.

We have one additional simplification for the data setup, where we ignore the remapped embedding of the first position, and the remapped embedding in the last position. We further simplify the setting by ignoring the remapped embedding in the first and last token, so the last position is deterministically $\phi(q)$ and the first position is $\phi(\epsilon_l)$ for some $l$. We now taxonomize the different types of tokens and get their corresponding probability over the two types of tasks:

For factual recalls, we have

- *Noisy Tokens*: Each $\phi(\epsilon_j)$ has a probability of $O\left(\frac{T}{N_n}\right)$ to be drawn for a single datapoint and a probability of $O\left(\frac{1}{N_n}\right)$ to share the same position with $\phi'(s)$.

- *Remapped Noisy Tokens*: Each $\phi'(\epsilon_j)$ has a probability of $O\left(\frac{T}{N_n}\right)$ to be drawn once and a probability of $O(\frac{1}{N_n})$ to share the same position with $\phi(s)$.

- *Subject Token and Remapped Subject Token*: By Assumption G.3, each $\phi(s)$ and $\phi'(s)$ must appear only once in full-batch gradient descent.

- *Query Token and Remapped Query Token*: $\phi(q)$ is deterministically fixed to be the last token for each datapoint. There are no $\phi'(q)$ in the factual recall task.

For induction, we have

- *Selected Noisy Token and Remapped Selected Noisy Token*: By Assumption G.3, each $\phi(\epsilon_j)$ will be selected as answer token only once in a full-batch gradient descent; so does $\phi'(\epsilon_j)$.

- *Trigger Token and Remapped Trigger Token*: $\phi(q)$ is deterministically to appear twice: one before the selected noisy token $\phi(\epsilon_j)$, and the other to be the EOS token. $\phi'(q)$ is guaranteed to share the same position with the answer token $\phi(\epsilon_j)$.

- *Unselected Noisy Token and Remapped Unselected Noisy Token*: Each token $\phi(\epsilon_k)$ has a probability of $O(\frac{T}{N_n})$ to be drawn for datapoint that it is not the answer and a probablity of $O\left(\frac{1}{N_n}\right)$ to share the same position with $\phi'(\epsilon_j)$. Their remapped embedding $\phi'(\epsilon_k)$ has a probablity of $O\left(\frac{T}{N_n}\right)$ to be drawn for datapoint that $\phi(\epsilon_k)$ is not the answer.

- *Factual Token and Remapped Factual Token*: $\phi(s)$ and $\phi'(s)$ will not appear in the induction dataset.

We will examine the signal of each token after the gradient steps.

**In the first step,** since we initialize both weight matrices to be zero, we have

$$\sigma_j = \frac{1}{T} \ \forall j \text{ and } \beta_k = \begin{cases} -\frac{1}{N} & \text{if } k \neq y \\ \frac{N-1}{N} & \text{if } k = y \end{cases} \text{ and } -\nabla_{W_{\mathrm{KQ}}}\mathcal{L} = 0 \tag{20}$$

This means we are essentially only optimizing $-\nabla_{W_{\mathrm{OV}}}\mathcal{L}$. For each datapoint in the factual recall dataset, suppose the factual token and its answer are $(s, y)$, we have that

$$\frac{\eta}{n}\mathbb{E}\left[\mu\left(y\right)^{\top}\left(-\nabla_{W_{\mathrm{OV}}}\mathcal{L}\right)\phi(s)\right] = O\left(\frac{\eta}{n}\cdot\beta_y\cdot\sigma_j\right) = O\left(\frac{\eta}{nT}\right) \tag{21}$$

$$\frac{\eta}{n}\mathbb{E}\left[\mu\left(y\right)^{\top}\left(-\nabla_{W_{\mathrm{OV}}}\mathcal{L}\right)\phi'(s)\right] = O\left(\frac{\eta}{nT}\right) \tag{22}$$

$$\frac{\eta}{n}\mathbb{E}\left[\mu\left(y\right)^{\top}\left(-\nabla_{W_{\mathrm{OV}}}\mathcal{L}\right)\phi(\epsilon_k)\right] = O\left(\frac{\eta}{nT}\cdot\frac{T}{N_n}\right) - \underbrace{O\left(\frac{1}{N}\cdot\frac{1}{T}\cdot\frac{T}{N_n}\cdot\frac{\eta}{n}\cdot n_F\right)}_{\text{fact incorrect terms}} - \underbrace{O\left(\frac{1}{N}\cdot\frac{1}{T}\cdot\frac{T}{N_n}\cdot\frac{\eta}{n}\cdot n_I\right)}_{\text{induction set}} \tag{23}$$

$$= O\left(\frac{\eta}{nN_n}\right) - O\left(\frac{\eta n_F}{NN_n n}\right) - O\left(\frac{\eta n_I}{NN_n n}\right) \tag{24}$$

$$= O\left(\frac{\eta}{nN_n}\right) - O\left(\frac{\eta}{NN_n}\right) \tag{25}$$

$$\frac{\eta}{n}\mathbb{E}\left[\mu\left(y\right)^{\top}\left(-\nabla_{W_{\mathrm{OV}}}\mathcal{L}\right)\phi'(\epsilon_k)\right] = O\left(\frac{\eta}{nN_n}\right) - O\left(\frac{\eta}{NN_n}\right) \tag{26}$$

$$\frac{\eta}{n}\mathbb{E}\left[\mu\left(y\right)^{\top}\left(-\nabla_{W_{\mathrm{OV}}}\mathcal{L}\right)\phi(q)\right] = O\left(\frac{\eta}{nT}\right) - \underbrace{O\left(\frac{\eta}{NT}\right)}_{\text{fact incorrect terms}} - \underbrace{O\left(\frac{2\eta}{NT}\right)}_{\text{induction set}} \tag{27}$$

where we can see that the most signal is absored in $\phi(s)$ with spurious correlations learned with $\phi'(s)$. The $W_{\mathrm{OV}}$ could act as associative-memory module for the factual recall dataset essentially in single gradient step. For other umembedding vector other than $\mu(y)$ to dot product with $(-\nabla_{W_{\mathrm{OV}}}\mathcal{L})\phi(\cdot)$, we remark that the gradient update from the factual recall dataset gives a negative value.

Take an arbitrary point in the induction dataset, suppose the selected answer token is $\epsilon_j$, we have

$$\frac{\eta}{n}\mathbb{E}\left[\mu\left(\epsilon_j\right)^{\top}\left(-\nabla_{W_{\mathrm{OV}}}\mathcal{L}\right)\phi\left(\epsilon_j\right)\right] = O\left(\frac{\eta}{nT}\right) - \underbrace{O\left(\frac{\eta}{NN_n}\right)}_{\text{fact and induction}} \tag{28}$$

$$\frac{\eta}{n}\mathbb{E}\left[\mu\left(\epsilon_j\right)^{\top}\left(-\nabla_{W_{\mathrm{OV}}}\mathcal{L}\right)\phi'\left(\epsilon_j\right)\right] = O\left(\frac{\eta}{nT}\right) - \underbrace{O\left(\frac{\eta}{NN_n}\right)}_{\text{fact and induction}} \tag{29}$$

$$\frac{\eta}{n}\mathbb{E}\left[\mu\left(\epsilon_j\right)^{\top}\left(-\nabla_{W_{\mathrm{OV}}}\mathcal{L}\right)\phi'\left(q\right)\right] = O\left(\frac{\eta}{nT}\right) - \underbrace{O\left(\frac{\eta n_I}{NTn}\right)}_{\text{induction only}} \tag{30}$$

$$\frac{\eta}{n}\mathbb{E}\left[\mu\left(\epsilon_j\right)^{\top}\left(-\nabla_{W_{\mathrm{OV}}}\mathcal{L}\right)\phi\left(q\right)\right] = O\left(\frac{\eta}{nT}\right) - \underbrace{O\left(\frac{\eta}{NT}\right)}_{\text{fact and induction}} \tag{31}$$

$$\frac{\eta}{n}\mathbb{E}\left[\mu\left(\epsilon_j\right)^{\top}\left(-\nabla_{W_{\mathrm{OV}}}\mathcal{L}\right)\phi\left(\epsilon_k\right)\right] = O\left(\frac{\eta}{nN}\right) - \underbrace{O\left(\frac{\eta}{NN_n}\right)}_{\text{fact and induction}} \tag{32}$$

$$\frac{\eta}{n}\mathbb{E}\left[\mu\left(\epsilon_j\right)^{\top}\left(-\nabla_{W_{\mathrm{OV}}}\mathcal{L}\right)\phi'\left(\epsilon_k\right)\right] = O\left(\frac{\eta}{nN}\right) - \underbrace{O\left(\frac{\eta}{NN_n}\right)}_{\text{fact and induction}} \tag{33}$$

where we can see thaet the $W_{\mathrm{OV}}$ terms learns the correct association between each $\mu(\epsilon_j)$ and $\phi(\epsilon)$, with spurious correlation learned with $\phi'(\epsilon_j)$. We further remark that this $W_{\mathrm{OV}}$ alone is able to make perfect predictions when loss is still high. However, as training progresses, the benign signal from $\nabla_{W_{\mathrm{OV}}}\mathcal{L}$ could also enable $W_{\mathrm{KQ}}$ to focus on the critical tokens.

Nowe we focus on the **second gradient step**. Since now $W_{\mathrm{KQ}}$ is still a zero matrix, we have

$$\sigma_j = \frac{1}{T} \quad \forall\, j \tag{34}$$

However, for now $\beta_k$ doesn't have an order $O(N)$ difference for $k = y$ and $k \neq y$. Here the relative update signal for $\nabla_{W_{\mathrm{OV}}}\mathcal{L}$ still follows from the analysis in the first step, where the relative update of the correct signal still dominates, but by a smaller margin. With a sufficiently large $\eta$ in the second step, the training could converge. Now we focus on how the second step leads to the desired form of $W_{\mathrm{KQ}}$.

For the induction task, we show that the model will concentrate on the correct term $\phi'(q) + \phi(\epsilon_j)$. Let's recall the gradient with respect to $W_{\mathrm{KQ}}$:

$$-\nabla_{W_{\mathrm{KQ}}}\mathcal{L} = \sum_{i=1}^{T} \gamma_i x_i x_T^T \qquad\qquad \left( \sum_{k=1}^{N} \beta_k \mu_k^\top W_{\mathrm{OV}} x_i \right) = \gamma_i$$

There are mainly "six types of inputs" in a single datapoint with selected answer token $\epsilon_j$: (1) desired focused term $\phi(\epsilon_j) + \phi'(q)$, (2) first occurrence of question $\phi(q) + \phi'(\epsilon_{j-2})$, (3) last position $\phi(q)$, (4) first position $\phi(\epsilon_1)$, (5) remapped answer token with unrelated noise $\phi(\epsilon_{j+1}) + \phi'(\epsilon_j)$, and (6) purely unrelated noise tokens $\phi(\epsilon_k) + \phi'(\epsilon_{k-1})$.

We claim that

$$\mathbb{E}\left[\gamma_j\right] > \mathbb{E}\left[\gamma_k\right] \tag{35}$$

where $j$ is the coefficient for the desired term (1) $\phi(\epsilon_j) + \phi'(q)$ and $k$ is any other types of terms. We can decompose

$$\gamma_i = \underbrace{\sum_{k \neq y}^{N} \beta_k \mu(k)^\top W_{\mathrm{OV}} x_i}_{\text{Small}} + \underbrace{\left(\beta_y \mu(y)^\top W_{\mathrm{OV}} x_i\right)}_{\text{Large}} \tag{36}$$

where now the subscript $y$ refers to the token $\epsilon_j$. We remark that the second term donimates the signal. From the analysis of the first gradient step, we know that

$$\mathbb{E}\left[\mu(y)^\top W_{\mathrm{OV}} \phi'(q)\right] > \mathbb{E}\left[\mu(y)^\top W_{\mathrm{OV}} \phi(\epsilon_j)\right] = \mathbb{E}\left[\mu(y)^\top W_{\mathrm{OV}} \phi'(\epsilon_j)\right] > \mathbb{E}\left[\mu(y)^\top W_{\mathrm{OV}}(\cdot)\right]$$

where $(\cdot)$ represents other terms (i.e., $\phi(q), \phi(\epsilon_k), \phi'(\epsilon_k)$). This means the term $\phi(\epsilon_j) + \phi'(q)$ has the largest signal ($\gamma_j$) in expectation. To see this, as we know $\beta_y > 0, \beta_k < 0$, consider substitute $\phi'(q)$ with any other terms (e.g. $\phi(q), \phi(\epsilon_k)$), then $\gamma_j$ is guaranteed to decrease. The same reasoning applies to $\phi(\epsilon_j)$ as we fix $\phi'(q)$. The only exception occurs with $\phi'(\epsilon_j)$, but we know that this term is guaranteed to not share the same position with $\phi'(q)$. Therefore, we finish our claim.

A similar statement can be made for the factual recall task where $W_{\mathrm{KQ}}$ concentrates on the $\phi(s) + \phi'(\epsilon_{i-1})$ and $\phi'(s) + \phi(\epsilon_{i+1})$ terms. The second term could be regarded as "benign spurious correlation" under our setup. We can take a sufficiently large $\eta$ in the second step to enable the convergence in expectation. As such, the $W_{\mathrm{KQ}}$ also takes in the form of Equation (8).

$\square$

**Corollary G.7** (Knowledge Conflict). *Under the knowledge conflict inference setting, the model capable of solving both* factual recall *and* induction *from Proposition 5.2 may output either the inductive token or the factual token. More specifically, if* $\exp(C_1)C_3 < \exp(C_2)C_4$, *then the model outputs the factual recall answer* $\mathcal{G}^*(s)$; *otherwise, the model outputs the induction answer* $\epsilon_j$.

*Proof.* The attention weight on the $\phi'(q) + \phi(\epsilon_j)$ is approximately $\frac{\exp(C_1)}{\exp(C_1)+\exp(C_2)+(T-2)}$; The attention weight on the $\phi(s) + \phi'(\epsilon_{i-1})$ is approximately $\frac{\exp(C_2)}{\exp(C_1)+\exp(C_2)+(T-2)}$.

The raw logit value of $\epsilon_j$ is $C_3 \frac{\exp(C_1)}{\exp(C_1)+\exp(C_2)+(T-2)}$ and the raw logit value of $\mathcal{G}^*(s)$ is $C_4 \frac{\exp(C_2)}{\exp(C_1)+\exp(C_2)+(T-2)}$. Other terms have a small logit values. Therefore, if $\exp(C_1)C_3 < \exp(C_2)C_4$, then the model outputs the factual recall answer $\mathcal{G}^*(s)$. Otherwise, the model outputs the induction answer $\epsilon_j$ □

**Proposition G.8** (Effectiveness of JuICE). *Consider the model from Proposition 5.2 and the case when its* inductive *part dominates (i.e., $\exp(C_1)C_3 >> \exp(C_2)C_4$), then the intervention by JuNE/PH3 of deleting the two attention heads is not as effective as JuICE. In particular, in this case JuNE/PH3 does not result in the parametric answer, while JuICE does.*

*Proof.* First, we remark that both attention heads (of the construction) are "highly influential" attention heads. As if one scales up or down the activation output of the two heads, the logit value of the corresponding parametric answer decreases or increases monotonically.

We now choose the intervention method to be knocking out for simplicity (which is exactly PH3; for JuICE, JuNE, it means adds a scaled version of the activation output by a factor of -1).

If we were using a single-pass intervention method advocated by PH3 or JuNE, then this simply means we delete the activation output from both heads, which gives an answer of random guessing among al elements in the vocabulary space $\mathcal{V}$.

If we use the dual-run design of JuICE, then we note that the activation outputs of the second layer from the first run has that

$$\text{Logit}_{fact}^{(1)} = \frac{C_4 \exp(C_2)}{\exp(C_1) + \exp(C_2) + (T-2)} \quad \text{Logit}_{ind}^{(1)} = \frac{C_3 \exp(C_1)}{\exp(C_1) + \exp(C_2) + (T-2)} \tag{37}$$

In the second run we have

$$\text{Logit}_{fact}^{(2)} = \frac{C_4 \exp(C_2)}{\exp(C_2) + (T-1)} \quad \text{Logit}_{ind}^{(2)} = \frac{C_3}{\exp(C_2) + (T-1)} \tag{38}$$

By deleting the activation output from the first run, we have

$$\text{Logit}_{fact}^{(2)*} > 0 \quad \text{Logit}_{ind}^{(2)*} < 0 \quad \text{Logit}_{other}^{(2)*} \approx 0 \tag{39}$$

This shows that JuICE results in the correct parametric answer. □

