# OpenReview forum: "Taming Knowledge Conflicts in Language Models"
_ICML.cc/2025/Conference — ICML 2025 spotlightposter_

### Official Review · Reviewer_xf53 · 2025-03-14

**Overall Recommendation:** 3

**Summary:**

This paper conducts a study of knowledge conflicts in LMs. The authors reveal that some attention heads contribute to both contextual information and parametric memory, instead of only contribute to one of them. They call this phenomenon "superposition of contextual information and parametric memory."

Besides these findings, the authors propose a training-free method for steering LMs towards either parametric or contextual knowledge, called JUICE (Just Run Twice).  JUICE first identifies consistent influential attention-head, then saves activations of these heads and at last steers the model by adding these saved activations.

Experimental results on many datasets show the effectiveness of JUICE. And the authors conduct theoretical analysis about the superposition phenomenon and the effectiveness of JUICE.

**Claims And Evidence:**

The claim about the superposition phenomenon is well supported by the analysis and experiments.
The effectiveness of the proposed method JUICE can be better analyzed.
At first, will changing the scaling factor of JUNE can obtain similar or superior performance than JUICE?
Besides, proposition 5.5 only analyzes the impact on JUNE after removing the attention head; however, in the actual experiment, a scaling method was used instead of actual removal. Therefore, compared to scaling, the benefits of adding the activations obtained from the first inference to steer still require further analysis.

**Essential References Not Discussed:**

No

**Experimental Designs Or Analyses:**

The experimental designs make sense.

**Methods And Evaluation Criteria:**

Yes, the evaluation criteria make sense.
The effectiveness of proposed steering method need further analysis.

**Other Comments Or Suggestions:**

See above.

**Other Strengths And Weaknesses:**

The paper is easy to understand. The proposed method achieved good performance in experiments. It would be better to add further analysis on why the steering approach is effective.

**Questions For Authors:**

Is the classification of these three types of knowledge conflicts comprehensive? If a new type of knowledge conflict emerges, can the proposed method be adapted to it?

**Relation To Broader Scientific Literature:**

The article points out the limitations of previous articles regarding the incentive explanation of knowledge conflicts and proposes the superposition phenomenon of attention heads concerning knowledge conflicts.

**Theoretical Claims:**

The theoretical claims about superposition phenomenon make sense.

---

> ### Author Rebuttal · Authors · 2025-03-30
>
> **Claim and Evidence 1: The Effect of Scaling JuNE**
>
> Thank you for the great suggestion. JuNE is an ablated variant of JuICE that only skips the second run but still incorporates and tunes its own scaling factor, as noted in Line 257 and Appendix E. We acknowledge that Proposition 5.5 may cause confusion, as JuNE is grouped with PH3 to emphasize its single-run nature, not the absence of scaling. We will clarify this distinction in the revised version of the paper.
>
> **Claim and Evidence 2: Further Analysis on Incorporating Activations from the First Run**
>
> This is indeed a great point! Proposition 5.5 is a proof-of-concept showing how dual-run methods improve intervention under simplified assumptions. It remains valid when using scaling factors rather than mere removal. In a high-level proof sketch, as the second layer transitions into a "factual memorizer" (due to the role alternation via scaling), single-pass methods continue to treat it as a context component, thereby suppressing its influence—whereas JuICE correctly preserves it, leading to more effective control.
>
> Beyond the theoretical discussion in Section 5, our paper provides extensive empirical support for this dual-run strategy. In Section 3, we introduce the observations that motivate the approach; we then validate its effectiveness in Tables 3 and 4 by comparing JuICE and JuNE, and we conduct a more detailed analysis in Figure 4. Collectively, these experiments and analyses offer strong evidence of the proposed method’s efficacy.
>
> Following your suggestion, we have also performed an **ablation study examining the benefits of adding the activations obtained from the first inference to steer (without scaling)** using Gemma on all parametric datasets. The details are provided in this [anonymous link](https://anonymous.4open.science/r/ICML_25_Rebuttal_Plot-82C6) (see xf53_Claim_and_Evidence_2). Simply adding these activations already confers considerable performance gains, demonstrating JuICE’s effectiveness. Applying scaling with well-chosen coefficients further amplifies these improvements. This also connects to Q2 raised by reviewer GwP9, where we observed that JuICE exhibits strong robustness to different scaling coefficients.
>
> **Question: Comprehensiveness of the classified knowledge conflicts and effectiveness of the proposed method on new conflicts**
>
> A primary contribution of our work is the in-depth exploration of language model mechanisms, accompanied by methods grounded in these insights. We classify three key types of knowledge conflicts, providing a clean and controlled setting that illuminates the internal workings of LMs—an aspect largely overlooked by prior research, which tends to focus on high-level behavioral observations. While we do not claim to cover every possible knowledge conflict, our classification addresses a wide range of conflicts and yields actionable insights. Building on the well-defined and representative cases explored in our work, future research can extend these insights to more atypical and complex forms of knowledge conflict.

---

### Official Review · Reviewer_GwP9 · 2025-03-14

**Overall Recommendation:** 3

**Summary:**

This paper proposes JuICE, a test-time intervention method to address knowledge conflicts in language models (LMs) by steering models toward either parametric or contextual knowledge. The method first identifies two sets of attention heads that consistently achieve the desired effect with either positive or negative scaling and then runs twice (saving outputs of the identified head and then scaling them).
Extensive experiments on 11 datasets and 6 model architectures demonstrate state-of-the-art performance and robustness across conflict types. Theoretical analysis formalizes knowledge conflicts and justifies JuICE’s effectiveness.

**Claims And Evidence:**

Yes

**Essential References Not Discussed:**

The paper did not cite IRCAN [1], which proposed to identify and reweight context-aware neurons to mitigate knowledge conflicts. This is very relevant to the topic and method in this paper and should be considered as a baseline.

[1] IRCAN: Mitigating Knowledge Conflicts in LLM Generation via Identifying and Reweighting Context-Aware Neurons.  NeurIPS 2024

**Experimental Designs Or Analyses:**

1. The comparison to the Prompt baseline is unfair. Prompt works significantly better for larger models (e.g. 70b) while the models in the experiments are no larger than 8b.

2. Limited Real-World Testing: Experiments focus on synthetic or curated conflicts; real-world noisy or partial conflicts (e.g., mixed relevant/irrelevant context) are not explored.

3. There are not enough baselines. For instance, one important baseline is missing as mentioned below.

**Methods And Evaluation Criteria:**

The proposed method is sound. The evaluation is comprehensive and makes sense.

**Other Comments Or Suggestions:**

Typos: In Table 3, 4 and many other places, "Llamma" -> "Llama"

The titles on the top of every page is missing.

**Other Strengths And Weaknesses:**

Strengths:
1. The paper provides novel insights: identifying CP superposition and challenging the prior assumption of mutually exclusive "memory heads" and "context heads."
2. JuICE requires no fine-tuning, no massive examples to identify target heads, and achieves consistent high performance across diverse domains, making it both lightweight and applicable.

Weaknesses: see above

**Questions For Authors:**

1. How does JuICE affect the general abilities of LLMs? Will it make the performance drop significantly on benchmarks such as MMLU, SuperGLUE?

2. How sensitive is JuICE to the choice of K (number of intervened heads) and scaling factors?

3. Can JuICE scale effectively to larger models (e.g., LLaMA 70B) with thousands of attention heads?

**Relation To Broader Scientific Literature:**

The paper proposes a novel method to mitigate knowledge conflicts and demonstrate that highly influential attention heads may simultaneously contribute to both memory and context, which could be very useful for future studies on knowledge conflicts.

**Theoretical Claims:**

N/A

---

> ### Author Rebuttal · Authors · 2025-03-30
>
> We sincerely thank the reviewer for the time and thoughtful feedback. Before addressing specific points, we emphasize that our primary goal is to understand the interplay between parametric (internal) and contextual (external) knowledge in LLMs. Our methodology primarily aims to validate the novel discovery, rather than exhaust engineering optimizations from all possible criteria, which, while having their own values, we see as orthogonal and suitable for future work.
>
> **Q1: General Abilities and Benchmarking LLMs with JuICE**
>
> JuICE is a targeted control tool that depends on whether the user intends to emphasize (a) parametric knowledge or (b) contextual information. Because to obtain the correct answer, such an intent varies across examples and remains an unknown prior in general benchmarks—even within the same dataset (i.e. question1 needs a while 2 needs b)—naively applying the same scaling factor to all examples can negatively affect performance. Consequently, JuICE is most effective when user intent is pre-defined, rather than for general-purpose benchmarking.
>
> **Q2: Sensitivity to Head Number and Scaling Factor**
>
> We provide the result of our sensitivity analysis in this [anonymous link](https://anonymous.4open.science/r/ICML_25_Rebuttal_Plot-82C6) (see GwP9_Q2). We plot the number of heads $K$ (from 2 to 30) and scaling factors $\alpha$ (grid search over 0-5) against the average test accuracy (of three conflict types) with Gemma on World Capital. The result shows that JUICE is highly robust against the choice of hyperparameters.
>
> **Q3: Scalability to Larger Models**
>
> JuICE runs linearly in the number of attention heads. Systems capable of running a 70B model can also accommodate JuICE.
>
> **Experimental Design 2: Real-world Conflict Complexity**
>
> We agree that complex cases (e.g., partial conflicts) are important but currently unexplored. We have explicitly acknowledged this limitation in Appendix F of the original paper. Before diving into this more complex problem, the task of understanding conflicts between parametric memory and contextual information (which is the focus of this paper) poses a **standalone and substantial** challenge, which has not been fully addressed by existing works. Our study employs datasets drawn from diverse domains and explores both parametric and contextual perspectives, presenting the first rigorous, comprehensive analysis of memory-context knowledge conflict. Tackling mixed or partial conflicts likely requires a dedicated study, which we leave for future work.
>
> **Experimental Designs 3 & 1: Baseline Comparison & Prompting**
>
> Thank you for highlighting this relevant work. IRCAN is indeed one of the best RAG-based hallucination methods focusing on one-sided knowledge conflicts. We will add it to the citation list. The problem setup of IRCAN and our work has many differences, and there exist multiple practical constraints (we will explain later on in detail). Accordingly, a point-to-point complete comparison between the two works is infeasible. However, we managed to identify a shared experimental setting between our paper and IRCAN (the Proverb-Ending dataset, referred to as MemoTrap in IRCAN). Using data reported in the IRCAN paper, we compare the two methods as follows:
>
> | Datasets | Proverb-Ending (MemoTrap in IRCAN) |
> | -------- | :-------: |
> | Gemma-2-IRCAN-CAD | 27.1 |
> | Gemma-2-JuICE | 74.6 |
> | LlaMA-2-IRCAN-CAD | 61.8 |
> | LlaMA-2-JuICE | 77.1 |
> | LlaMA-3-IRCAN-CAD | 54.4 |
> | LlaMA-3-JuICE | 75.4 |
>
> JuICE consistently outperforms IRCAN while requiring smaller identification sets. We will add this comparison to the revision.
>
> Here we further explain the infeasibility of a point-to-point comparison between these two methods:
> - Universal Component Discovery: IRCAN uses dataset-specific splits; it does not claim cross-domain generality. For our Table 4, we require either a domain-agnostic or no identification set to ensure fair comparison across tasks.
> - Heavy Tuning: IRCAN requires ~288 hyperparameter configurations (a 16×18 search grid), plus a large validation set (same size as test). Our benchmarks (e.g., NQ-swap, 5k samples) make this computationally infeasible within the rebuttal window.
>
> Prompting Baseline: The reviewer notes prompt baselines work better for large models (e.g., 70B). Unfortunately, our computing resources cannot even load such models. Our chosen model scales match or exceed those used in prior high-impact interpretability papers [1–4], including IRCAN. Consistent with prior work (e.g., PH3), we adopt prompting as our baseline.
>
> [1] Locating and Editing Factual Associations in GPT, NeurIPS'22 (1143 citations)
>
> [2] Inference-Time Intervention, NeurIPS'23 (409 citations)
>
> [3] Improving Alignment and Robustness with Circuit Breakers, NeurIPS'24 (61 citations)
>
> [4] Cutting Off the Head Ends the Conflict, ACL'24 (Baseline)
>
>
> **Typos**
>
> Thank you for your careful checking. We will amend these typos and format issues in the revised version of our paper.

---

### Official Review · Reviewer_bLHn · 2025-03-17

**Overall Recommendation:** 3

**Summary:**

The author proposed JUICE (Just Run Twice), a test-time intervention method to address knowledge conflicts in language models (LMs), where parametric memory (internal knowledge) contradicts contextual information. Extensive experiments across 11 datasets and 6 model architectures demonstrate that JUICE achieves state-of-the-art performance and robust generalization. For example, under the Gemma-2b model, JUICE consistently outperforms baselines like PH3 and Prompt, achieving high accuracy even in challenging coherent conflict scenarios. The method also shows strong robustness against hyperparameter variations and paraphrased inputs.

**Claims And Evidence:**

Yes

**Essential References Not Discussed:**

No

**Experimental Designs Or Analyses:**

Yes.

**Methods And Evaluation Criteria:**

Yes

**Other Comments Or Suggestions:**

1. It might be helpful to provide visual examples of how model outputs change before and after applying JUICE, especially in challenging conflict scenarios.

**Other Strengths And Weaknesses:**

Strengths
1. Both experimental results and theoretical analysis are provided.

2. Improvements are achieved, which shows the advantages of the proposed method.


Weaknesses
1. Dual-run inference will increase the running time to get the final results.

**Questions For Authors:**

1. JUICE's effectiveness  has been shown on several models. How do you expect it to perform on models with significantly different architectures, such as those using sparse attention?

2.  The superposition of contextual information and parametric memory arises naturally from the training process. How can this insight be leveraged to design models that inherently avoid such conflicts, rather than relying on post-hoc interventions like JUICE?

**Relation To Broader Scientific Literature:**

The proposed might be use in many domains since knowledge conflict is very common.

**Theoretical Claims:**

The reviewer did not check the proofs.

---

> ### Author Rebuttal · Authors · 2025-03-30
>
> **Q1: Effectiveness of JuICE on different architectures**
>
> Thank you for the insightful question. Our main paper evaluates JuICE on a range of widely used dense attention-based open-source LMs. Following your suggestion, we additionally test JuICE on Mistral-v0.1-7B, a leading sparse attention-based model. Results below confirm that JuICE remains highly effective in this architecture, reinforcing the generality of our findings:
>
> | Dataset Conflict | World-Capital Clean | World-Capital Subs-conf | World-Capital Coh-conf |
> |------------------|:-------------------:|:-----------------------:|:----------------------:|
> | Original         |         96.2        |           76.8          |           2.7          |
> | Prompt           |         96.2        |           81.1          |           7.0          |
> | PH3$_l$           |         95.7        |           92.4          |          23.8          |
> | PH3$_s$            |         96.2        |           85.9          |          20.0          |
> | JuICE            |         96.1        |         **93.0**        |        **92.4**        |
>
>
> | Dataset Conflict | Official-Language Clean | Official-Language Subs-conf | Official-Language Coh-conf |
> |------------------|:-------------------:|:-----------------------:|:----------------------:|
> | Original         |         96.2        |           38.9          |           0.0          |
> | Prompt           |         96.2        |           54.3          |           0.6          |
> | PH3$_l$            |         95.0        |           72.8          |          0.6          |
> | PH3$_s$            |         94.4        |           43.8          |          0.0          |
> | JuICE            |         96.3       |         **90.1**        |        **82.5**        |
>
> | Dataset Conflict | Company-Founder Clean | Company-Founder Subs-conf | Company-Founder Coh-conf |
> |------------------|:-------------------:|:-----------------------:|:----------------------:|
> | Original         |         59.3        |           8.8          |           0.0          |
> | Prompt           |         59.3        |           23.1          |           1.6          |
> | PH3$_l$            |         54.9        |           20.3          |          0.0         |
> | PH3$_s$            |         59.3        |           8.2          |         0.0          |
> | JuICE            |        57.7       |         **44.5**        |        **17.3**        |
>
> These results show that JuICE retains strong performance in sparse-attention models. We will include them in the revised version.
>
> **Q2: Insights on Model Design**
>
> Thank you for bringing up this excellent point. At present, there may be no definitive answer to whether superposition is inherently "harmful", which would require a new, independent investigation. On the one hand, superposition can impede interpretability by making it difficult to disentangle the roles of individual parameters. On the other hand, we speculate it may enhance a model’s compactness and expressiveness, allowing the same parameters to accomplish multiple tasks. The reviewer's comment has inspired us to come up with a bolder speculation: this phenomenon might shed lights on understanding the success of some relatively small models (such as QwQ-32b), which has demonstrated remarkable reasoning performance on par with much bigger ones (such as Deepseek-r1-671b) with much more compact parameter size.
>
> In our paper, we propose methods that recognize the inevitability of superposition, a perspective that has received little attention so far. We would like to leave further investigation to future works: how the LLM components accommodating superposition contribute to model expressiveness and whether increasing such components can lead to more compact models with equal expressiveness.
>
> **Weakness: Additional Runtime**
>
> We agree that JuICE's strong performance comes with modest runtime overhead, as acknowledged in Appendix F. Future work may improve its efficiency. Additionally, users can trade off speed and performance by choosing JuNE, our lightweight variant, which runs faster and still performs on par with or better than other baselines.
>
> **Other Comment: Providing Visual Examples of Model Output Changes**
>
> Thank you for the excellent suggestion! We now visualize how JuICE shifts model output probabilities across conflicting scenarios. For each case, we plot the average probability mass over the parametric token, contextual token, and other tokens (calculated as 1 - P(parametric) - P(contextual)) on the World Capital dataset using Gemma. The plot is available at this [anonymous link](https://anonymous.4open.science/r/ICML_25_Rebuttal_Plot-82C6) (see bLHn_comment). The results clearly show JuICE consistently promotes the desired parametric token across all conflict types. We plan to include this plot in the revised version.

---

### Official Review · Reviewer_HLU5 · 2025-03-18

**Overall Recommendation:** 4

**Summary:**

The authors contribute a new method and analysis framework for understanding knowledge conflicts in RAG applications. Specifically, they define the usual setting of knowledge conflict where the contextual (retrieved) knowledge may differ from the parametric (internal) knowledge of the LLM. In these cases, and active area of research is understanding how RAG agents choose between the contextual and the parametric knowledge.

The authors contribute the following:

0. introduction of the concept of CP (contextual-parametric) superposition, which is the idea that depending on the input tokens, the LLM will lean more toward parameteric or external knowledge
1. through a weight-masking study, a motivation for controlling knowledge conflict tradeoff via scaling attention head outputs
2. a methodology that tunes an attention-head scaler via examining intermediate outputs of the model
3. a comprehensive empirical study showing that the proposed method (JUICE) is singularly most effective at controlling knowledge conflict tradeoff
4. theoretical characterization of the underpinnings of JUICE, showing that a LLM can indeed achieve CP superposition via pretraining (thus justifying the overall approach to this problem)

**Claims And Evidence:**

Yes, I found all claims to be convincing.

**Essential References Not Discussed:**

No essential references I could find that were missed

**Experimental Designs Or Analyses:**

I checked all experimental designs presented in the paper. No issues.

**Methods And Evaluation Criteria:**

Yes, the authors define the problem they study as CP superposition, namely the ability of models to choose between parametric and internal knowledge. The problem is therefore not to be the most factual, but rather to influence a certain outcome with high probability. The authors' empirical study is well-crafted around this objective.

**Other Comments Or Suggestions:**

Very minor nit: there may be some confusion with Algorithm 1, Step One: Save Important Streams: $H_l \in S_1$ doesn't quite make sense, because $H_l$ appears to be an output, but $S_1$ is a parameter set of the model

**Other Strengths And Weaknesses:**

Nothing else to report here

**Questions For Authors:**

No further questions

**Relation To Broader Scientific Literature:**

In my opinion, this paper brings a strong empirical and theoretical understanding to the field of knowledge conflicts and more broadly LLM factuality. The vast majority of existing work has conducted only empirical studies that expose the behavior of RAG agents under knowledge conflict scenarios. This is one of the only works I'm aware of to actually propose a well-motivated and effective intervention.

**Theoretical Claims:**

I did not check the correctness of the proofs, but I read through the main results in detail, and they are quite coherent.

---

> ### Author Rebuttal · Authors · 2025-03-30
>
> Thank you for your thoughtful feedback and for recognizing the significance of our work. We agree with the reviewer that many of the ideas presented in our paper extend beyond the scope of knowledge conflicts, as superposition appears to be a widespread phenomenon. Gaining a deeper understanding of its underlying mechanism will ultimately lead to more robust and reliable use of language models.
>
> Additionally, we really appreciate your careful checking and attention to detail in our manuscript. In the revised version, we will clarify the notations in Algorithm 1 according to your recommendations.

---

### Decision · Program_Chairs · 2025-05-01

**Decision:**

Accept (spotlight poster)

**Comment:**

The paper puts forward a superposition perspective on knowledge conflicts where model components use both parameteric and context information simultaneously. They also provide a new method to steer models by making two passes through the model, using the first pass to identify various categories of attention heads and get reliable directions for steering. The paper is generally well-written, has clear experiments, baselines, theoretical analyses. The paper is a clear accept!